# A causal role for right frontopolar cortex in directed, but not random, exploration

Wojciech K Zajkowski[1], Malgorzata Kossut[2,3], Robert C Wilson[4,5]*

[1]University of Social Sciences and Humanities, Warsaw, Poland; [2]Department of Psychology, University of Social Sciences and Humanities, Warsaw, Poland; [3]Nencki Institute of Experimental Biology, Warsaw, Poland; [4]Department of Psychology, University of Arizona, Tucson, United States; [5]Cognitive Science Program, University of Arizona, Tucson, United States

**Abstract** The explore-exploit dilemma occurs anytime we must choose between exploring unknown options for information and exploiting known resources for reward. Previous work suggests that people use two different strategies to solve the explore-exploit dilemma: directed exploration, driven by information seeking, and random exploration, driven by decision noise. Here, we show that these two strategies rely on different neural systems. Using transcranial magnetic stimulation to inhibit the right frontopolar cortex, we were able to selectively inhibit directed exploration while leaving random exploration intact. This suggests a causal role for right frontopolar cortex in directed, but not random, exploration and that directed and random exploration rely on (at least partially) dissociable neural systems.
DOI: https://doi.org/10.7554/eLife.27430.001

*For correspondence:
bob@arizona.edu

Competing interests: The authors declare that no competing interests exist.

## Introduction

In an uncertain world adaptive behavior requires us to carefully balance the exploration of new opportunities with the exploitation of known resources. Finding the optimal balance between exploration and exploitation is a hard computational problem and there is considerable interest in understanding how humans and animals strike this balance in practice (*Badre et al., 2012*; *Cavanagh et al., 2011*; *Cohen et al., 2007*; *Daw et al., 2006*; *Frank et al., 2009*; *Hills et al., 2015*; *Mehlhorn et al., 2015*; *Wilson et al., 2014*). Recent work has suggested that humans use two distinct strategies to solve the explore-exploit dilemma: directed exploration, based on information seeking, and random exploration, based on decision noise (*Wilson et al., 2014*). Even though both of these strategies serve the same purpose, that is, balancing exploration and exploitation, it is likely they rely on different cognitive mechanisms. Directed exploration is driven by information and is thought to be computationally complex (*Gittins and Jones, 1979*; *Auer et al., 2002*; *Gittins, 1974*). On the other hand, random exploration can be implemented in a simpler fashion by using neural or environmental noise to randomize choice (*Thompson, 1933*).

A key question is whether these dissociable behavioral strategies rely on dissociable neural systems. Of particular interest is the frontopolar cortex (FPC) – an area that has been associated with a number of functions, such as tracking pending and/or alternate options (*Koechlin and Hyafil, 2007*; *Boorman et al., 2009*), strategies (*Domenech and Koechlin, 2015*) and goals (*Pollmann, 2016*) and that has been implicated in exploration itself (*Badre et al., 2012*; *Cavanagh et al., 2011*; *Daw et al., 2006*). Importantly, however, the exact role that FPC plays in exploration is unknown as how exploration is defined varies from paper to paper. In one line of work, exploration is defined as information seeking. Understood this way, exploration correlates with RFPC activity measured via fMRI (*Badre et al., 2012*) and a frontal theta component in EEG (*Cavanagh et al., 2011*), suggesting a role for RFPC in directed exploration. However, in another line of work, exploration is

operationalized differently, as choosing the low value option, not the most informative. Such a measure of exploration is more consistent with random exploration where decision noise drives the sampling of low value options by chance. Defined in this way, exploratory choice correlates with lateral FPC activation (*Daw et al., 2006*) and stimulation and inhibition of RFPC with direct current (tDCS) can increase and decrease the frequency with which such exploratory choices occur (*Raja Beharelle et al., 2015*).

Taken together, these two sets of findings suggest that RFPC plays a crucial role in both directed and random exploration. However, we believe that such a conclusion is premature because of a subtle confound that arises between reward and information in most explore-exploit tasks. This confound arises because participants only gain information from the options they choose, yet are incentivized to choose more rewarding options. Thus, over many trials, participants gain more information about more rewarding options and the two ways of defining exploration, that is, choosing high information or low reward options, become confounded (*Wilson et al., 2014*). This makes it impossible to tell whether the link between RFPC and exploration is specific to either directed or random exploration, or whether it is general to both.

To distinguish these interpretations and investigate the causal role of FPC in directed and random exploration, we used continuous theta-burst TMS (*Huang et al., 2005*) to selectively inhibit right frontopolar cortex (RFPC) in participants performing the 'Horizon Task', an explore-exploit task specifically designed to separate directed and random exploration (*Wilson et al., 2014*). Using this task we find evidence that inhibition of RFPC selectively inhibits directed exploration while leaving random exploration intact.

## Results

We used our previously published 'Horizon Task' (*Figure 1*) to measure the effects of TMS stimulation of RFPC on directed and random exploration. In this task, participants play a set of games in which they make choices between two slot machines (one-armed bandits) that pay out rewards from different Gaussian distributions. To maximize their rewards in each game, participants need to exploit the slot machine with the highest mean, but they cannot identify this best option without exploring both options first.

The Horizon Task has two key manipulations that allow us to measure directed and random exploration. The first manipulation is the horizon itself, i.e. the number of decisions remaining in each game. The idea behind this manipulation is that when the horizon is long (6 trials), participants should explore more frequently, because any information they acquire from exploring can be used to make better choices later on. In contrast, when the horizon is short (1 trial), participants should exploit the option they believe to be best. Thus, this task allows us to quantify directed and random exploration as changes in information seeking and behavioral variability that occur with horizon.

The second manipulation is the amount of information participants have about each option *before* making their first choice. This information manipulation is achieved by using four forced-choice trials, in which participants are told which option to pick, at the start of each game. We use these forced-choice trials to setup one of two information conditions: an unequal, or (*Aston-Jones and Cohen, 2005*; *Badre et al., 2012*), condition, in which participants see 1 play from one option and 3 plays from the other option, and an unequal, or (*Auer et al., 2002*; *Auer et al., 2002*), condition, in which participants see two outcomes from both options. By varying the amount of information participants have about each option *independent* of the mean payout of that option, this information manipulation allows us to remove the reward-information confound, at least on the first free-choice trial (*Figure 2*). After the first free-choice trial, however, participants tend to choose more rewarding options more frequently and reward and information are rapidly confounded. For this reason the bulk of our analyses are focussed on the first free-choice trial where the confound has been removed.

### RFPC stimulation selectively inhibits directed exploration on the first free-choice

In this section we analyze behavior on the first free-choice trial in each game. This way we are able to remove any effect of the reward-information confound and fairly compare behavior between horizon conditions. We analyze the data with both a model-free approach, using simple statistics of the

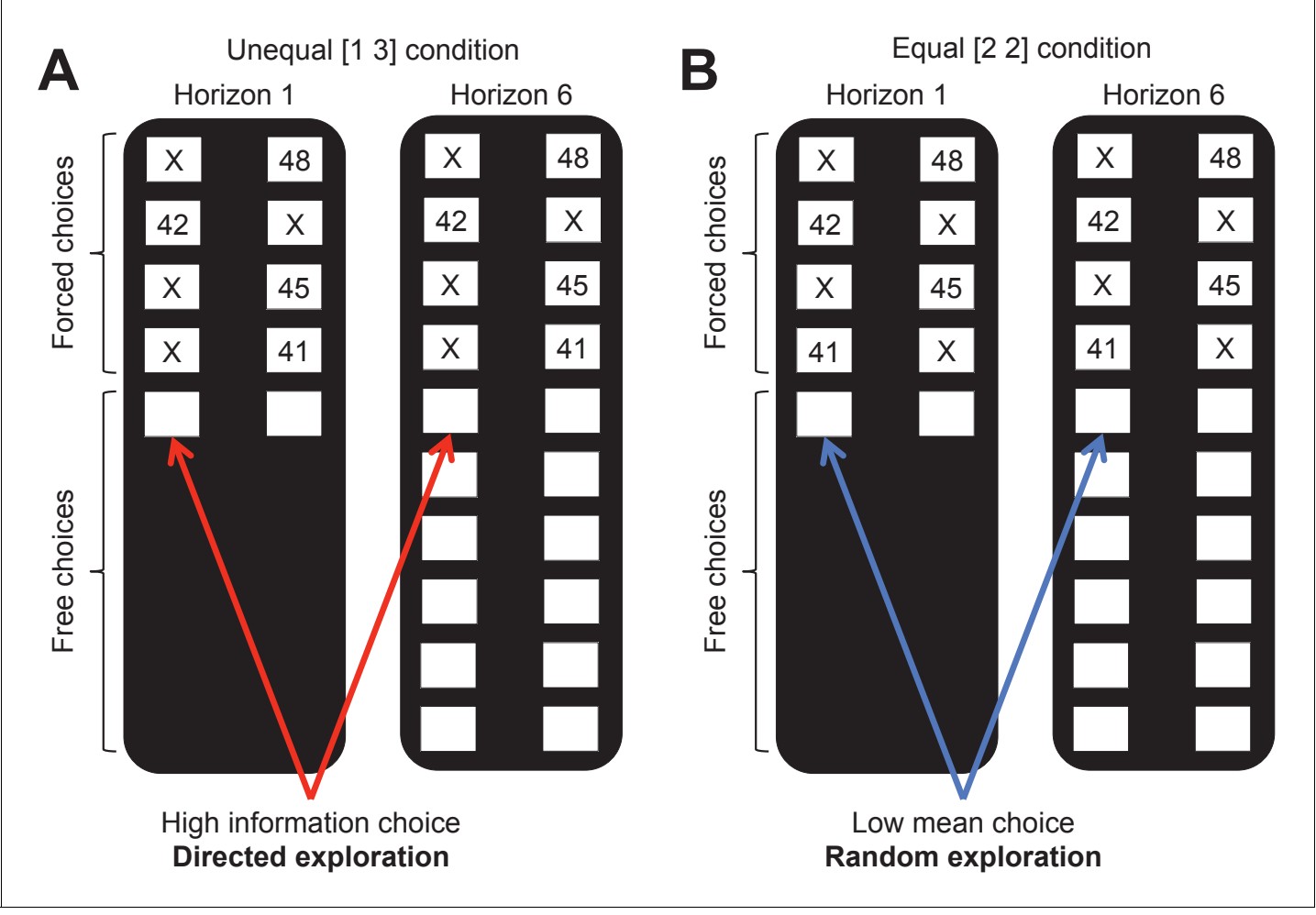

**Figure 1.** The horizon task. Participants make a series of decisions between two one-armed bandits that pay out probabilistic rewards with unknown means. At the start of each game, 'forced-choice' trials give participants partial information about the mean of each option. We use the forced-choice trials to set up one of two information conditions: (**A**) an unequal (or [1 3]) condition in which participants see 1 play from one option and 3 plays from the other and (**B**) an equal (or [2 2]) condition in which participants see 2 plays from both options. A model-free measure of directed exploration is then defined as the change in information seeking with horizon in the unequal condition (**A**). Likewise a model-free measure of random exploration is defined as the change choosing the low mean option in the equal condition (**B**).
DOI: https://doi.org/10.7554/eLife.27430.002

data to quantify directed and random exploration, as well as a model-based approach, using a cognitive model of the behavior to draw more precise conclusions. Both analyses point to the same conclusion that RFPC stimulation selectively inhibits directed, but not random, exploration.

## Model-free analysis

The two information conditions in the Horizon Task allow us to quantify directed and random exploration in a model-free way. In particular, directed exploration, which involves information seeking, can be quantified as the probability of choosing the high information option, $p(\text{high info})$ in the [1 3] condition, while random exploration, which involves decision noise, can be quantified as the probability of making a mistake, or choosing the low mean reward option, $p(\text{low mean})$, in the [2 2] condition.

Using these measures of exploration, we found that inhibiting the RFPC had a significant effect on directed exploration but not random exploration (**Figure 3A,B**). In particular, for directed exploration, a repeated measures ANOVA with horizon, TMS condition and order as factors revealed a significant interaction between stimulation condition and horizon ($F(1, 24) = 4.96$, $p = 0.036$).

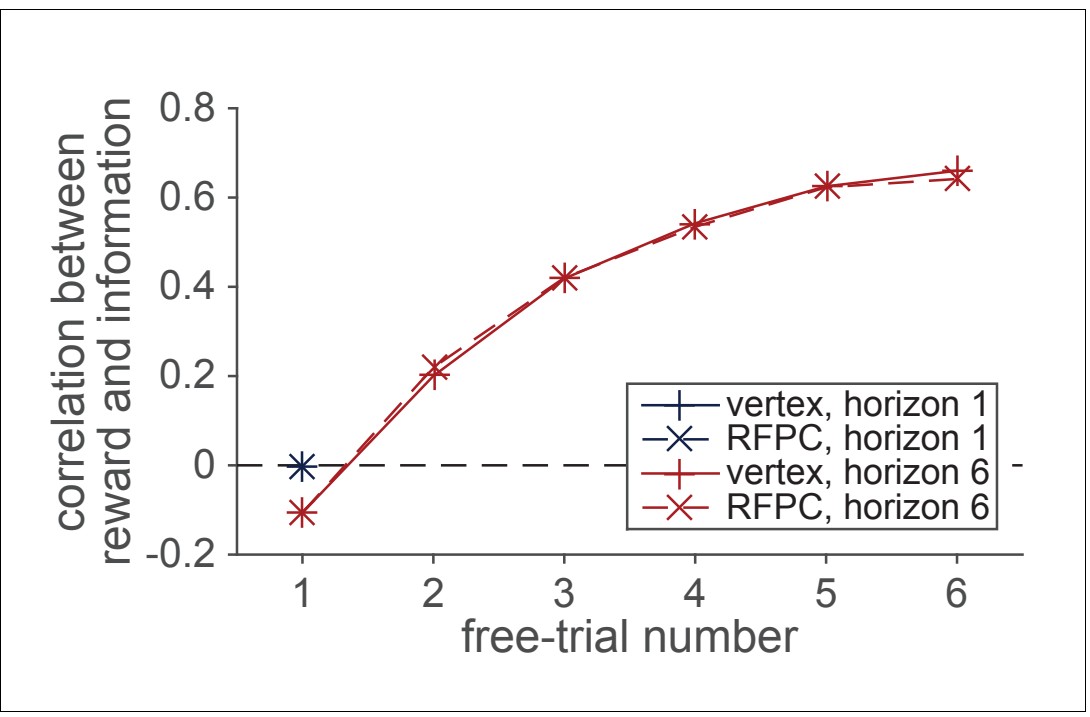

**Figure 2.** The reward-information confound. The y-axis corresponds to the correlation between the sign of the difference in mean ($\mathrm{sgn}(\mu_{left} - \mu_{right})$) between options and the sign of difference in the number of times each option has been played ($\mathrm{sgn}(n_{left} - n_{right})$). The forced trials are chosen such that the the correlation is approximately zero on the first free-choice trial. After the first trial, however, a positive correlation quickly emerges as participants choose the more rewarding options more frequently. This strong confound between reward and information makes it difficult to dissociate directed and random exploration on later trials.
DOI: https://doi.org/10.7554/eLife.27430.003

Conversely, a similar analysis for random exploration revealed no effects of stimulation condition (main effect of stimulation condition, $F(1, 24) = 0.88$, $p = 0.36$; interaction of stimulation condition with horizon, $F(1, 24) = 1.24$, $p = 0.28$). Post hoc analyses revealed that the change in directed exploration was driven by changes in information seeking in horizon 6 (one-sided t-test, $t(24) = 2.62$, $p = 0.008$) and not in horizon 1 (two-sided t-test, $t(24) = -0.30$).

## Model-based analysis

While the model-free analyses are intuitive, the model-free statistics, $p(\text{high info})$ and $p(\text{low mean})$, are not pure reflections of information seeking and behavioral variability and could be influenced by other factors such as spatial bias and learning. To account for these possibilities we performed a model-based analysis using a model that extends our earlier work (**Wilson et al., 2014**; **Somerville et al., 2017**; **Krueger et al., 2017**) see Materials and methods for a complete description. In this model, the level of directed and random exploration is captured by two parameters: an information bonus for directed exploration, and decision noise for random exploration. In addition the model includes terms for the spatial bias and to describe learning.

### Overview of model

Before presenting the results of the model-based analysis we begin with a brief overview of the most salient points of the model. A full description of the model can be found in the Methods and code to implement the model can be found in the Supplementary Material.

Conceptually, the model breaks the explore-exploit choice down into two components: a learning component, in which participants estimate the mean payoff of each option from the rewards they see, and a decision component, in which participants use this estimated payoff to guide their choice. The learning component assumes that participants compute an estimate of the average payoff for

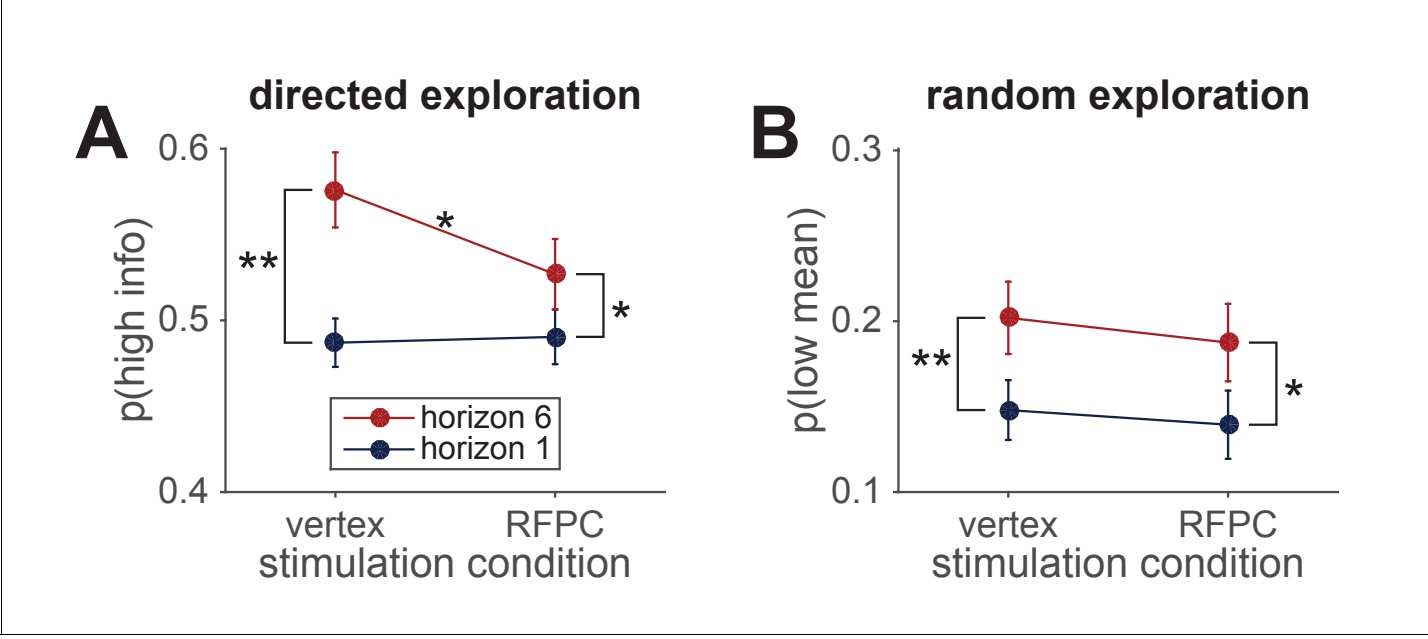

**Figure 3.** Model-free analysis of the first free-choice trial shows that RPFC stimulation affects directed, but not random, exploration. (**A**) In the control (vertex) condition, information seeking increases with horizon, consistent with directed exploration. When RFPC is stimulated, directed exploration is reduced, an effect that is entirely driven by changes in horizon 6 (* denotes $p<0.02$ and ** denotes $p<0.005$; error bars are $\pm$ s.e.m.). (**B**) Random exploration increases with horizon but is not affected by RFPC stimulation.

DOI: https://doi.org/10.7554/eLife.27430.004

each slot machine, $R_t^i$, using a simple delta rule update equation (based on a Kalman filter (**Kalman, 1960**), see Materials and methods):

$$R_{t+1}^i = R_t^i + \alpha_t^i(r_t - R_t^i) \tag{1}$$

where $r_t$ is the reward on trial $t$ and $\alpha_t^i$ is the time-varying learning rate that determines the extent to which the prediction error, $(r_t - R_t^i)$, updates the estimate of the mean of bandit $i$. The learning process is described by three free parameters: the initial value of the estimated payoff, $R_0$, and two learning rates, the initial learning rate, $\alpha_1$, and the asymptotic learning rate, $\alpha_{\inf}$, which together describe the evolution of the actual learning rate, $\alpha_t$, over time. For simplicity, we assume that these parameters are independent of horizon and uncertainty condition (**Table 1**).

**Table 1.** Model parameters.

Subject's behavior on the first free choice of each session is described by 13 free parameters. Three of these parameters ($R_0$, $\alpha_1$ and $\alpha_\infty$) describe the learning process and do not vary with horizon or uncertainty condition. Ten of these parameters ($A$, $B$ and $\sigma$ in the different horizon and information conditions) describe the decision process. All parameters are estimated for each subject in each stimulation condition and the key analysis asks whether parameters change between vertex and RFPC stimulation.

| Parameter | Horizon dependent? | Uncertainty dependent? | TMS dependent? |
|---|---|---|---|
| prior mean, $R_0$ | no | no | yes |
| initial learning rate, $\alpha_1$ | no | no | yes |
| asymptotic learning rate, $\alpha_\infty$ | no | no | yes |
| information bonus, $A$ | yes | n/a | yes |
| spatial bias, $B$ | yes | yes | yes |
| decision noise, $\sigma$ | yes | yes | yes |

DOI: https://doi.org/10.7554/eLife.27430.005

The decision component of the model assumes that participants choose between the two options (left and right) probabilistically according to.

$$p(\text{choose right}) = \frac{1}{1 + \exp\left(\frac{\Delta R + A\Delta I + B}{\sigma}\right)} \tag{2}$$

where $\Delta R$ ( $= R_t^{left} - R_t^{right}$ ) is the difference in expected reward between left and right options and $\Delta I$ is the difference in information between left and right options (which we define as +1 when left is more informative, −1 when right is more informative, and 0 when both options convey equal information in the [2 2] condition). The decision process is described by three free parameters: the information bonus $A$, the spatial bias $B$, and the decision noise $\sigma$. We estimate separate values of the decision parameters for each horizon and (since the information bonus is only used in the [1 3] condition) separate values of only the bias and decision noise for each uncertainty condition.

Overall, subject's behavior in each session (vertex vs RFPC stimulation) is described by 13 free parameters (*Table 1*): three describing learning ($R_0$, $\alpha_1$ and $\alpha_\infty$) and 10 describing the decision process ($A$ in the two horizon conditions, $B$ and $\sigma$ in the four horizon-x-uncertainty conditions). These 13 parameters were fit to each subject in each stimulation condition using a hierarchical Bayesian approach (*Lee and Wagenmakers, 2014*) (see Materials and methods).

## Model fitting results

Posterior distributions over the group-level means are shown in the left column of *Figure 4*, while posteriors over the TMS-related change in parameters are shown in the right column. Both columns suggest a selective effect of RFPC stimulation on the information bonus in horizon 6.

Focussing on the left column first, overall the parameter values seem reasonable. The prior mean is close to the generative mean of 50 used in the actual experiment, and the decision parameters are comparable to those found in our previous work (*Wilson et al., 2014*). The learning rate parameters, $\alpha_1$ and $\alpha_\infty$, were not included in our previous models and are worth discussing in more detail. As expected for Bayesian learning (*Kalman, 1960*; *Nassar et al., 2010*), the initial learning rate is higher than the asymptotic learning rate (95% of samples in the vertex condition, 94% in the RFPC condition). However, the actual values of the learning rates are quite far from their 'optimal' settings of $\alpha_1 = 1$ and $\alpha_\infty = 0$ that would correspond to perfectly computing the mean reward. This suggests a greater than optimal reliance on the prior ($\alpha_1 < 1$) and a pronounced recency bias ($\alpha_\infty > 0$) such that the most recent rewards are weighted more heavily in the computation of expected reward, $R_t^i$. Both of these findings are likely due to the fact that the version of the task we employed did not keep the outcomes of the forced trials on screen and instead relied on people's memories to compute the expected value.

Turning to the right hand column of *Figure 4*, we can see that the model-based analysis yields similar result to the model-free analysis. In particular we see a reduction (of about 4.8 points) in the information bonus in horizon 6 (with 99% of samples showing a reduced information bonus in the RFPC stimulation condition) and no effect on decision noise in either horizon in either the [2 2] or [1 3] uncertainty conditions (with between 40% and 63% of samples below zero).

In addition to the effect on the information bonus in horizon 6, there is also a hint of an effect on the information bonus in horizon 1 (85% samples less than zero) and on the prior mean $R_0$ (88% samples above zero). While these results may suggest that RFPC stimulation affects more than just information bonus in horizon 6, they more likely reflect an inherent tradeoff between prior mean and information bonus that is peculiar to this task. In particular, because the prior mean has a stronger effect on the more uncertain option, an increase in $R_0$ increases the value of the more informative option in much the same way as an information bonus. Thus, when applied to this task, the model has a built in tradeoff between prior mean and information bonus that can muddy the interpretation of both. Note that this tradeoff is not a general feature of the model and could be removed with a different task design that employed more forced choice trials and hence more time for the effects of the prior to be removed.

*Figure 5* exposes the tradeoff between $R_0$ and $A$ in more detail. Panels A and B plot samples from the posterior over the TMS-related change in information bonus, $A(\text{vertex}) - A(\text{RFPC})$, against the TMS-related change in prior mean, $R_0(\text{vertex}) - R_0(\text{RFPC})$. For both horizon conditions we see a strong negative correlation such that increasing $R_0$ decreases $A$. This negative correlation is

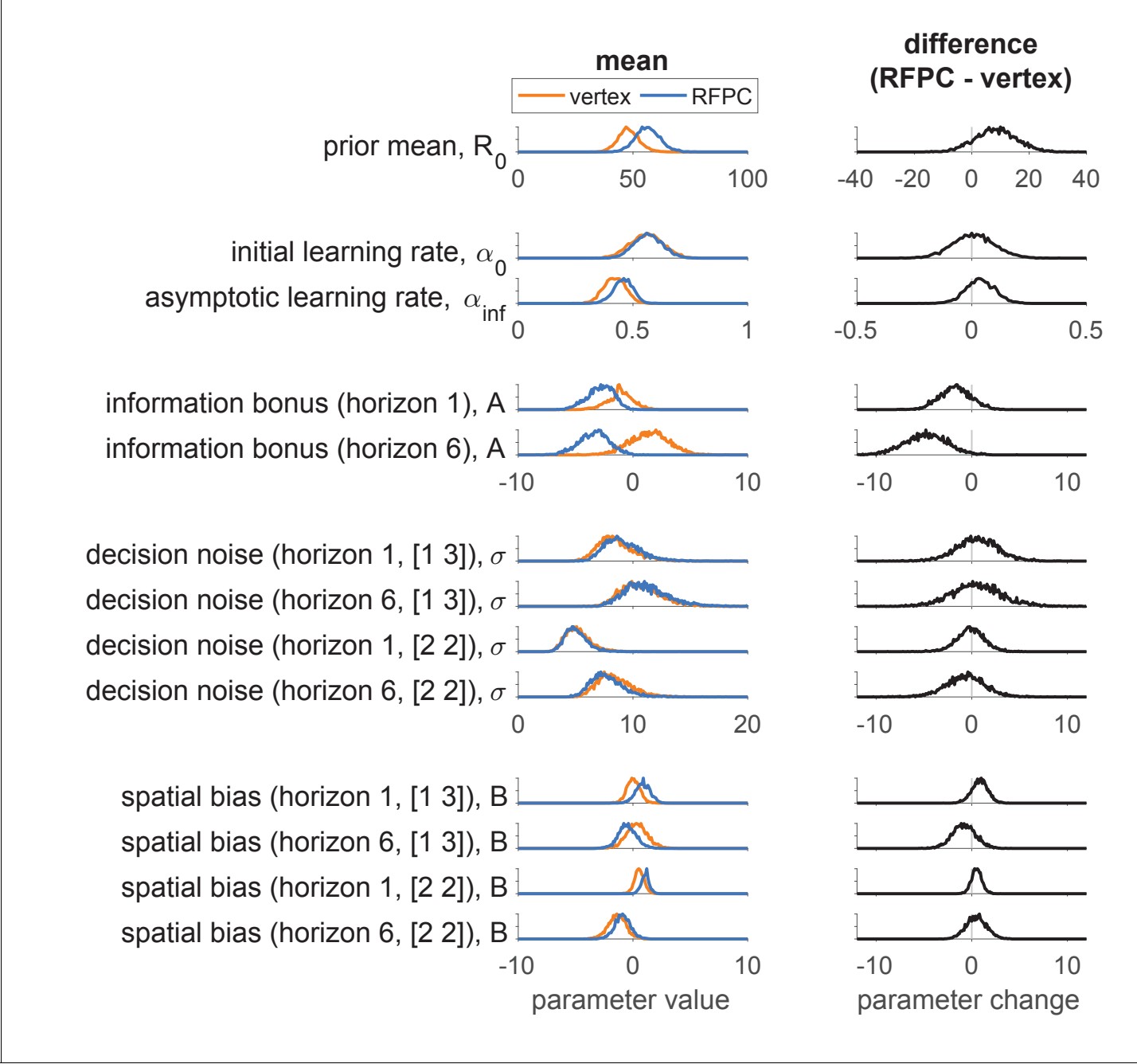

**Figure 4.** Model-based analysis of the first free-choice trial showing the effect of RFPC stimulation on each of the 13 parameters. Left column: Posterior distributions over each parameter value for RFPC and vertex stimulation condition. Right column: posterior distributions over the change in each parameter between stimulation conditions. Note that, because information bonus, decision noise and spatial bias are all in units of points, we plot them on the same scale to facilitate comparison of effect size.

DOI: https://doi.org/10.7554/eLife.27430.006

especially problematic for the interpretation of the horizon 1 change in information bonus where a sizable fraction of the posterior centers on no change in either variable. In contrast the negative correlation between $A$ and $R_0$ does not affect our interpretation of the horizon 6 result where the TMS-related change in $A$ is negative regardless of of the change in $R_0$.

Finally we asked whether the horizon-dependent change in information seeking, i.e. $\Delta A = A(h = 6) - A(h = 1)$, was different in each TMS condition. As shown in **Figure 5C**, the TMS-

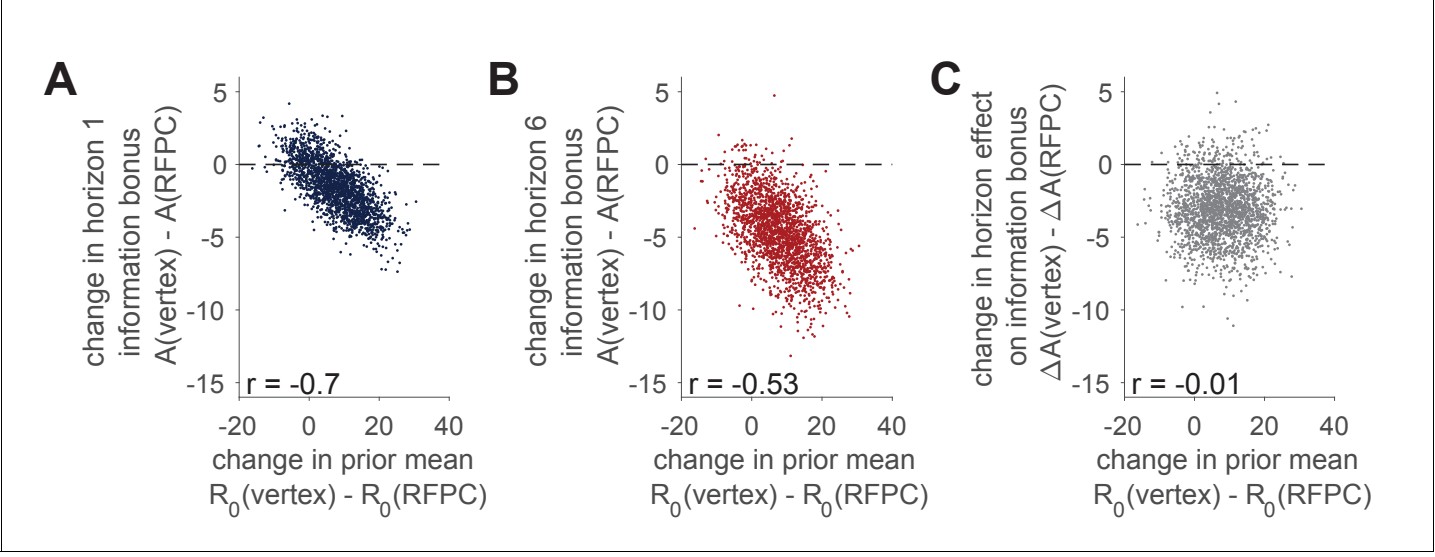

**Figure 5.** Correlation between TMS-induced changes in information bonus, $A$, and TMS-induced changes in the prior mean, $R_0$. (A, B) Samples from the posterior distributions over the TMS-related changes in prior mean, $R_0$, and TMS-related change in information bonus in horizon 1 (A) and horizon 6 (B). In both cases we see a negative correlation between the change in $R_0$ and the change in $A$ consistent with a tradeoff between these variables in the model. (C) Samples from the posterior over the effect of TMS stimulation on the horizon-related change in information bonus, $\Delta A = A(h = 6) - A(h = 1)$ plotted against samples from the TMS-related change in prior mean. Here we see no correlation between variables and the majority of $\Delta A(\text{vertex}) - \Delta A(\text{RFPC})$ samples below zero consistent with an effect of RFPC stimulation on directed exploration.
DOI: https://doi.org/10.7554/eLife.27430.007

related change in $\Delta A$ is about $-3.1$ points (94% samples below 0) and is uncorrelated with the TMS-related change in $R_0$. Taken together, this suggests that we can be fairly confident in our claim that RFPC stimulation has a selective effect on directed exploration.

## The effect of RFPC stimulation on later trials

Our analyses so far have focussed on just the first free choice and have ignored the remaining five choices in the horizon six games. The reason for this is the reward-information confound, illustrated in *Figure 2*, which makes interpretation of the later trials more difficult. Despite this difficulty, we note that in *Figure 2* the size of the confound is almost identical in the two stimulation conditions and so we proceed, with caution, to present a model-free analysis of the later trials below.

In *Figure 6* we plot the model-free measures, $p(\text{high info})$ and $p(\text{low mean})$, as a function of trial number. Both measures show a decrease over the course of the horizon six games although, because of the confound, it is difficult to say whether these changes reflect a reduction in directed exploration, random exploration, or both. Interestingly, the differences in $p(\text{high info})$ between vertex and RFPC conditions on the first free-choice trial appear to persist into the second, a result that becomes more apparent when we plot the TMS-related change, that is, $p(\text{high info}, \text{RFPC}) - p(\text{high info}, \text{vertex})$ (*Figure 6C,D*). More formally a repeated measures ANOVA with trial number, TMS condition as factors reveals a significant main effect of trial number ($F(5, 120) = 126$, $p < 10^{-45}$), no main effect of TMS condition ($F(1, 120) = 1.17$, $p = 0.29$) and a near significant interaction between trial number and TMS condition ($F(5, 120) = 2.26, p = 0.053$). A post hoc, one-sided t-test on the second trial reveals a marginally significant reduction in $p(\text{high info})$ on the second trial ($t(24) = 1.61$). In contrast, a similar analysis for random exploration shows no evidence for any effect of TMS condition on $p(\text{low mean})$ (main effect of TMS, $F(1, 120) = 0.16$, $p = 0.69$; TMS x trial number, $F(5, 120) = 0.69$, $p = 0.63$) although the main effect of trial number persists ($F(5, 120) = 13.7$, $p < 10^{-9}$). Thus, the analysis of later trials provides additional, albeit modest, support for the idea that RFPC stimulation selectively disrupts directed but not random exploration at long horizons.

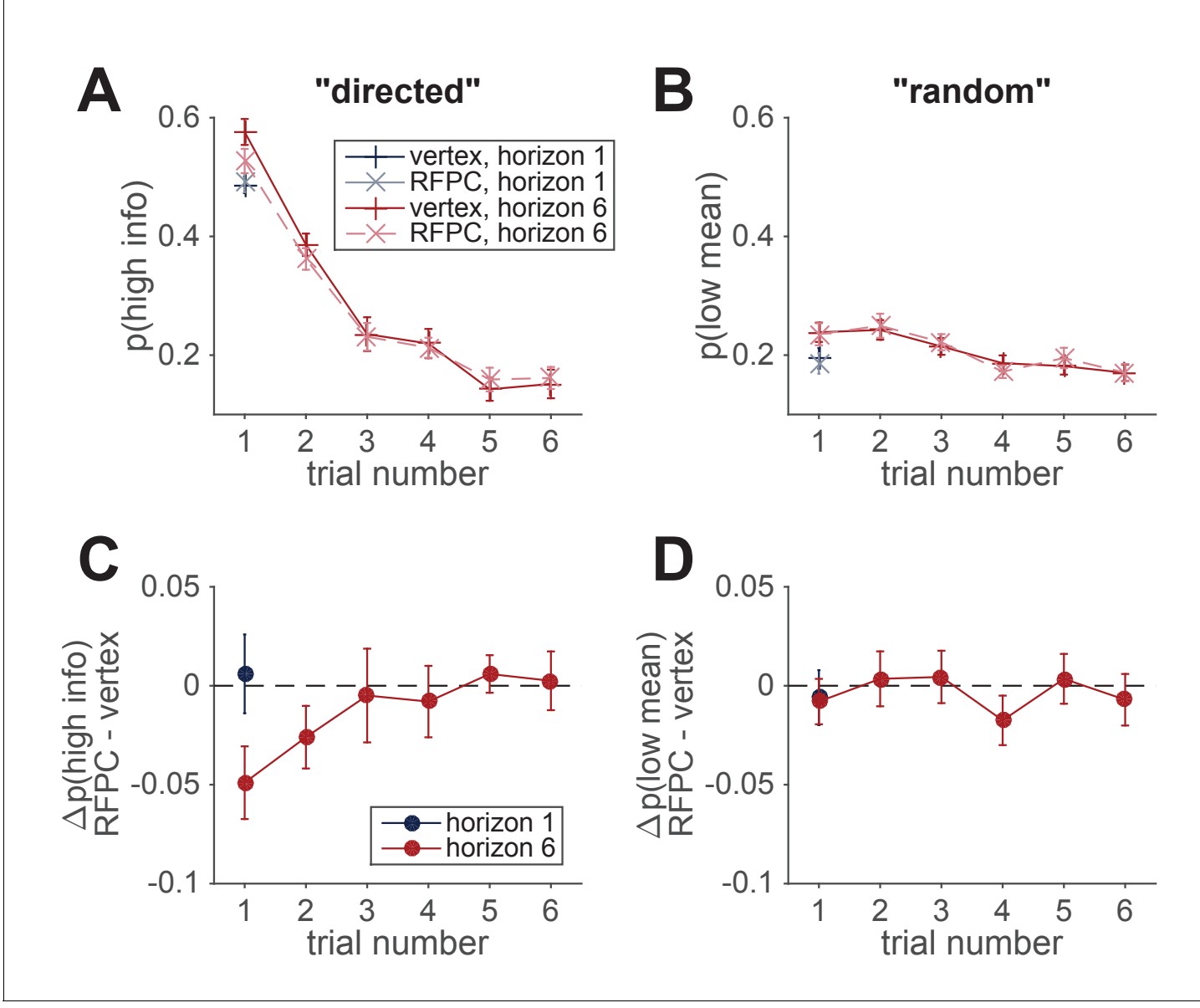

**Figure 6.** Model-free analysis of all trials. (**A**, **B**) Model-free measures of directed (**A**) and random (**B**) exploration as a function of trial number suggests a reduction in both directed and random exploration over the course of the game. (**C**, **D**) TMS-induced change in measures of directed and random exploration as a function of trial number. This suggests that the reduction in directed exploration on the first free-choice trial, persists into the second trial of the game.

DOI: https://doi.org/10.7554/eLife.27430.008

## Discussion

In this work we used continuous theta-burst transcranial magnetic stimulation (cTBS) to investigate whether right frontopolar cortex (RFPC) is causally involved in directed and random exploration. Using a task that is able to behaviorally dissociate these two types of exploration, we found that inhibition of RFPC caused a selective reduction in directed, but not random exploration. To the best of our knowledge, this finding represents the first causal evidence that directed and random exploration rely on dissociable neural systems and is consistent with our recent findings showing that directed and random exploration have different developmental profiles (*Somerville et al., 2017*). This suggests that, contrary to the assumption underlying many contemporary studies (*Daw et al., 2006*; *Badre et al., 2012*), exploration is not a unitary process, but a dual process in which the

distinct strategies of information seeking and choice randomization are implemented via distinct neural systems.

Such a dual-process view of exploration is consistent with the classical idea that there are multiple types of exploration (**Berlyne, 1966**). In particular Berlyne's constructs of 'specific exploration', involving a drive for information and 'diversive exploration', involving a drive for variety, bear a striking resemblance to our definitions of directed and random exploration. Despite the importance of Berlyne's work, more modern views of exploration tend not to make the distinction between different types of exploration, considering instead a single exploratory state or exploratory drive that controls information seeking across a wide range of tasks (**Berlyne, 1966**; **Aston-Jones and Cohen, 2005**; **Hills et al., 2015**; **Kidd and Hayden, 2015**). At face value, such unitary accounts seem at odds with a dual-process view of exploration. However, these two viewpoints can be reconciled if we allow for the possibility that, while directed and random exploration are implemented by different systems, their levels are set by a common exploratory drive.

Intriguingly, individual differences in behavior on the Horizon Task provide some support for the idea that directed and random exploration are driven by a common source. In particular, in a large behavioral data set of 277 people performing the Horizon Task, we find a positive correlation between the levels of directed and random exploration such that people with high levels of directed exploration also tend to have high levels random exploration ($r(275) = 0.29$, $p<10^{-5}$), **Figure 7**. This is consistent with the idea that the levels of directed and random exploration are set by the strength of an exploratory drive that varies as an individual difference between people.

While the present study does allow us to conclude that directed and random exploration rely on different neural systems, the limited spatial specificity of TMS limits our ability to say exactly what those systems are. In particular, because the spatial extent of TMS is quite large, stimulation aimed

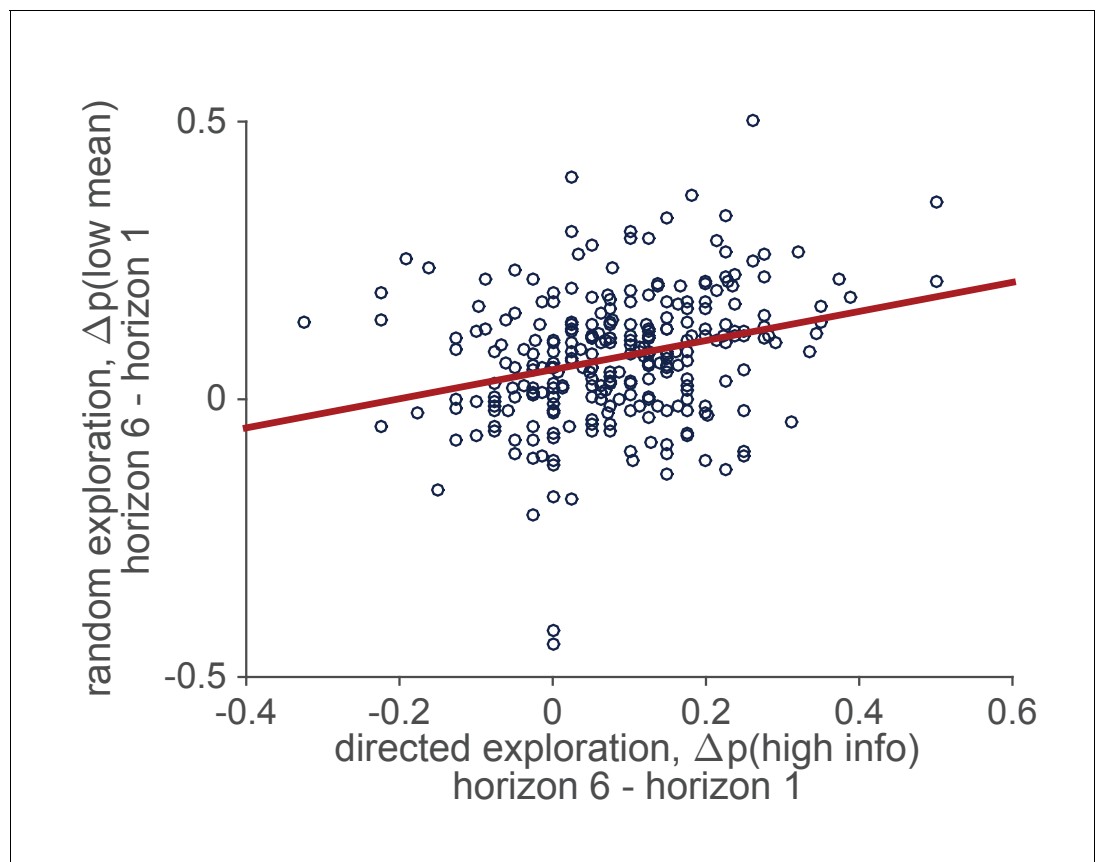

**Figure 7.** Correlation between individual differences in the levels of directed and random exploration in a sample of 277 people performing the Horizon Task.

DOI: https://doi.org/10.7554/eLife.27430.009

at frontal pole may directly affect activity in nearby areas such as ventromedial prefrontal cortex (vmPFC) and orbitofrontal cortex (OFC), both areas that have been implicated in exploratory decision making and that may be contributing to our effect (*Daw et al., 2006*). In addition to such direct effects of TMS on nearby regions, indirect changes in areas that are connected to the frontal pole could also be driving our effect. For example, cTBS of left frontal pole has been associated with changes in blood perfusion in areas such as amygdala, fusiform gyrus and posterior parietal cortex (*Volman et al., 2011*) and by changes in the fMRI BOLD signal in OFC, insula and striatum (*Hanlon et al., 2017*). In addition (*Volman et al., 2011*) showed that unilateral cTBS of left frontal pole is associated with changes in blood perfusion to the right frontal pole. Indeed, such a bilateral effect of cTBS may explain why our intervention was effective at all given that a number of neuroimaging studies have shown bilateral activation of the frontal pole associated with exploration (*Daw et al., 2006*; *Badre et al., 2012*). Future work combining cTBS with neuroimaging will be necessary to shed light on these issues.

With the above caveats that our results may not be entirely due to disruption of frontal pole, the interpretation that RFPC plays a role in directed, but not random, exploration is consistent with a number of previous findings. For example, frontal pole has been associated with tracking the value of the best unchosen option (*Boorman et al., 2009*), inferring the reliability of alternate strategies (*Boorman et al., 2009*; *Domenech and Koechlin, 2015*), arbitrating between old and new strategies (*Donoso et al., 2014*; *Mansouri et al., 2015*), and reallocating cognitive resources among potential goals in underspecified situations (*Pollmann, 2016*). Taken together, these findings suggest a role for frontal pole in model-based decisions (*Daw et al., 2006*) that involve long-term planning and the consideration of alternative actions. From this perspective, it is perhaps not surprising that directed exploration relies on RFPC, since computing an information bonus relies heavily on an internal model of the world. It is also perhaps not surprising that random exploration is independent of RFPC, as this simpler strategy could be implemented without reference to an internal model. Indeed, the ability to explore effectively in a model-free manner, may be an important function of random exploration as it allows us to explore even when our model of the world is wrong.

More generally, it is unlikely that frontal pole is the only area involved in directed exploration, and more work will be needed to map out the areas involved in directed and random exploration and expose their causal relationship to explore-exploit behavior.

## Materials and methods

### Participants

31 healthy right-handed, adult volunteers (19 female, 12 male; ages 19–32). An initial sample size of 16 was chosen based on two studies using a very similar cTBS design that stimulated lateral FPC (*Costa et al., 2011*; *Costa et al., 2013*) and this was augmented to 31 on the basis of feedback from reviewers. Five participants (5 female, 0 male) were excluded from the analysis due to chance-level performance in both experimental sessions. One (female) participant failed to return for the second (vertex stimulation condition) session and is excluded from the model-free analyses but not the model-based analyses as that can handle missing data more gracefully. Thus our final data set consisted of 25 participants (13 female, 12 male, ages 19–32) with complete data and one participant (female, aged 20) with data from the RFPC session only.

All participants were informed about potential risks connected to TMS and signed a written consent. The study was approved by University of Social Sciences and Humanities ethics committee.

### Procedure

There were two experimental TMS sessions and a preceding MRI session. On the first session T1 structural images were acquired using a 3T Siemens TRIO scanner. The scanning session lasted up to 10 min. Before the first two sessions, participants filled in standard safety questionnaires regarding MRI scanning and TMS. During the experimental sessions, prior to the stimulation participants went through 16 training games to get accustomed to the task. Afterwards, resting motor thresholds were obtained and the stimulation took place. Participants began the main task immediately after stimulation. The two experimental sessions were performed with an intersession interval of at least 5

days. The order of stimulation conditions was counterbalanced across subjects. All sessions took place at Nencki Institute of Experimental Biology in Warsaw.

## Stimulation site

The RFPC peak was defined as [x,y,z]= [35,50,15] in MNI (Montreal Neurological Institute) space. The coordinates were based on a number of fMRI findings that indicated RFPC involvement in exploration (*Badre et al., 2012*; *Boorman et al., 2009*; *Daw et al., 2006*) and constrained by the plausibility of stimulation (e.g. defining 'z' coordinate lower would result in the coil being placed uncomfortably close to the eyes). Vertex corresponded to the Cz position of the 10–20 EEG system. In order to locate the stimulation sites we used a frameless neuronavigation system (Brainsight software, Rogue Research, Montreal, Canada) with a Polaris Vicra infrared camera (Northern Digital, Waterloo, Ontario, Canada).

## TMS protocol

We used continuous theta burst stimulation (cTBS) (*Huang et al., 2005*). cTBS requires 50 Hz stimulation at 80% resting motor threshold. 40 s stimulation is equivalent to 600 pulses and can decrease cortical excitability for up to 50 min (*Wischnewski and Schutter, 2015*).

Individual resting motor thresholds were assessed by stimulating the right motor knob and inspecting if the stimulation caused an involuntary hand twitch in 50% of the cases. We used a Mag-Pro X100 stimulator (MagVenture, Hueckelhoven, Germany) with a 70 mm figure-eight coil. The TMS was delivered in line with established safety guidelines (*Rossi et al., 2009*).

## Limitations

Defining stimulation target by peak coordinates based on findings from previous studies did not allow to account for individual differences in either brain anatomy or the impact of TMS on brain networks (*Gratton et al., 2013*). However, a study by Volman and colleagues (*Volman et al., 2011*) that used the same theta-burst protocol on the left frontopolar cortex has shown bilateral inhibitory effects on blood perfusion in the frontal pole. This suggests that both right and left parts of the frontopolar cortex might have been inhibited in our experiment, which is consistent with imaging results indicating bilateral involvement of the frontal pole in exploratory decisions.

## Task

The task was a modified version of the Horizon Task (*Wilson et al., 2014*). As in the original paper, the distributions of payoffs tied to bandits were independent between games and drawn from a Gaussian distribution with variable means and fixed standard deviation of 8 points. Participants were informed that in every game one of the bandits was objectively 'better' (has a higher payoff mean). Differences between the mean payouts of the two slot machines were set to either 4, 8, 12 or 20. One of the means was always equal to either 40 or 60 and the second was set accordingly. The order of games was randomized. Mean sizes and order of presentation were counterbalanced. Participants played 160 games and the whole task lasted between 39 and 50 min (mean 43.4 min).

Each game consisted of 5 or 10 choices. Every game started with a screen saying 'New game' and information about whether it was a long or short horizon, followed by sequentially presented choices. Every choice was presented on a separate screen, so that participants had to keep previous the scores in memory. There was no time limit for decisions. During forced choices participants had to press the prompted key to move to the next choice. During free choices they could press either 'z' or 'm' to indicate their choice of left or right bandit. The decision could not be made in a time shorter than 200 ms, preventing participants from accidentally responding too soon. The score feedback was presented for 500 ms. A counter at the bottom of the screen indicated the number of choices left in a given game. The task was programmed using PsychoPy software v1.86 (*Peirce, 2007*).

Participants were rewarded based on points scored in two sessions. The payoff bounds were set between 50 and 80 zl (equivalent to approximately 12 and 19 euro). Participants were informed about their score and monetary reward after the second session.

Finally, the random seeds were not perfectly controlled between subjects. The first 16 subjects ran the task with identical random seeds and thus all 16 saw the same sequence of forced-choice

trials in both vertex and RFPC sessions. For the remaining subjects the random seed was unique for each subject and each session, thus these subjects had unique a series of forced-choice trials for each session. Despite this limitation we saw no evidence of different behavior across the two groups.

## Data and code

Behavioral data as well as Matlab code to recreate the main figures from this paper can be found on the Dataverse website at https://dataverse.harvard.edu/dataset.xhtml?persistentId=doi:10.7910/DVN/CZT6EE.

## Model-based analysis

We modeled behavior on the first free choice of the Horizon Task using a version of the logistic choice model in *Wilson et al. (2014)* that was modified to include a learning component. In particular, we assume that participants use the outcomes of the forced-choice trials to learn an estimate of the mean reward of each option, before inputting that mean reward into a decision function that includes terms for directed and random exploration. This model naturally decomposes into a learning component and a decision component and we consider each of these components in turn.

### Learning component

The learning component of the model assumes that participants use a Kalman filter to learn a value for the mean reward of each option. The Kalman filter (*Kalman, 1960*) has been used to model learning in other explore-exploit tasks (*Daw et al., 2006*) and is a popular model of Bayesian learning as it is both analytically tractable and easily relatable to the delta-rule update equations of reinforcement learning.

More specifically, the Kalman filter assumes a generative model in which the rewards from each bandit, $r_t$, are generated from Gaussian distribution with a fixed standard deviation, $\sigma_r$, and a mean, $m_t^i$, that is different for each bandit and can vary over time. The time dependence of the mean is determined by a Gaussian random walk with mean 0 and standard deviation $\sigma_d$. Note that this generative model, assumed by the Kalman filter, is slightly different to the true generative model used in the Horizon Task, which assumes that the mean of each bandit is constant over time, that is, $\sigma_d = 0$. This mismatch between the assumed and actual generative models, is quite deliberate and allows us to account for the suboptimal learning of the subjects. In particular, this mismatch, introduces the possibility of a recency bias (when $\sigma_d > 0$) whereby more recent rewards are over-weighted in the computation of $R_t^i$.

The actual equations of the Kalman filter model are straightforward. The model keeps track of an estimate of both the mean reward, $R_t^i$, of each option, $i$, and the uncertainty in that estimate, $\sigma_t^i$. When option $i$ is played on trial $t$, these two parameters update according to

$$R_{t+1}^i = R_t^i + \frac{(\sigma_{t+1}^i)^2}{\sigma_r^2}(r_t - R_t^i) \tag{3}$$

and

$$\frac{1}{(\sigma_{t+1}^i)^2} = \frac{1}{(\sigma_t^i)^2 + \sigma_d^2} + \frac{1}{\sigma_r^2} \tag{4}$$

When option i is not played on trial t we assume that the estimate of the mean stays the same, but that the uncertainty in this estimate grows as the generative model assumes the mean drifts over time. Thus for unchosen option $j$ we have

$$R_{t+1}^j = R_t^j \quad \text{and} \quad (\sigma_{t+1}^j)^2 = (\sigma_t^j)^2 + \sigma_d^2 \tag{5}$$

When the option is played, the update *Equation 3* for $R_t^i$ is essentially just a 'delta rule' (*Rescorla and Wagner, 1972*; *Schultz et al., 1997*), with the estimate of the mean being updated in proportion to the prediction error, $r_t - R_t^i$. This relationship to the reinforcement learning literature is made more clear by rewriting the learning equations in terms of the time varying learning rate,

$$\alpha_t^i = \frac{(\sigma_{t+1}^i)^2}{\sigma_r^2} \tag{6}$$

Written in terms of this learning rate, *Equations 3 and 4* become

$$R_{t+1}^i = R_t^i + \alpha_t^i(r_t - R_t^i) \tag{7}$$

and

$$\frac{1}{\alpha_t^i} = \frac{1}{\alpha_{t-1}^i + \alpha_d} + 1 \tag{8}$$

where

$$\alpha_d = \frac{\sigma_d^2}{\sigma_r^2} \tag{9}$$

The learning model has four free parameters, the noise variance, $\sigma_r^2$, the drift variance, $\sigma_d^2$, and the initial values of the estimated reward, $R_0$, and uncertainty in that variance estimate, $\sigma_0^2$. In practice, only three of these parameters are identifiable from behavioral data, and we will find it useful to reparameterize the learning model in terms of $R_0$ and an initial, $\alpha_1$, and asymptotic, $\alpha_\infty$, learning rate. In particular, the initial value of the learning rate relates to $\sigma_0$, $\sigma_r$ and $\sigma_d$ as

$$\alpha_1 = \frac{\sigma_0^2 + \sigma_d^2}{\sigma_r^2} \tag{10}$$

While the asymptotic value of the learning rate, which corresponds to the steady state value of $\alpha_t^i$ if option $i$ is played forever, relates to $\alpha_d$ (and hence $\sigma_d$ and $\sigma_r$) as

$$\alpha_\infty = \frac{1}{2}\left(-\alpha_d + \sqrt{\alpha_d^2 + 4\alpha_d}\right) \tag{11}$$

While this choice to parameterize the learning equations in terms of $\alpha_1$ and $\alpha_\infty$ is somewhat arbitrary, we feel that the learning rate parameterization has the advantage of being slightly more intuitive and leads to parameter values between 0 and 1 which are (at least for us) easier to interpret.

### Decision component

Once the payoffs of each option, $R_t^i$, have been estimated from the outcomes of the forced-choice trials, the model makes a decision using a simple logistic choice rule:

**Table 2.** Model parameters, priors, hyperparameters and hyperpriors.

| Parameter | Prior | Hyperparameters | Hyperpriors |
|---|---|---|---|
| prior mean, $R_0^{\tau s}$ | $R_0^{\tau s} \sim$ Gaussian($\mu_{R_0}^\tau, \sigma_{R_0}^\tau$) | $\theta_{R_0}^\tau = (\mu_{R_0}^\tau, \sigma_{R_0}^\tau)$ | $\mu_{R_0}^\tau \sim$ Gaussian( 50, 14 ) <br> $\sigma_{R_0}^\tau \sim$ Gamma( 1, 0.001 ) |
| initial learning rate, $\alpha_1^{\tau s}$ | $\alpha_1^{\tau s} \sim$ Beta($a_{\alpha_1}^\tau, b_{\alpha_1}^\tau$) | $\theta_{\alpha_1}^\tau = (a_{\alpha_1}^\tau, b_{\alpha_1}^\tau)$ | $a_{\alpha_1}^\tau \sim$ Uniform( 0.1, 10 ) <br> $b_{\alpha_1}^\tau \sim$ Uniform( 0.5, 10 ) |
| asymptotic learning rate, $\alpha_\infty^{\tau s}$ | $\alpha_\infty^{\tau s} \sim$ Beta($a_{\alpha_\infty}^\tau, b_{\alpha_\infty}^\tau$) | $\theta_{\alpha_\infty}^\tau = (a_{\alpha_\infty}^\tau, b_{\alpha_\infty}^\tau)$ | $a_{\alpha_\infty}^\tau \sim$ Uniform( 0.1, 10 ) <br> $b_{\alpha_\infty}^\tau \sim$ Uniform( 0.1, 10 ) |
| information bonus, $A^{\tau shu}$ | $A^{\tau shu} \sim$ Gaussian($\mu_A^{\tau hu}, \sigma_A^{\tau hu}$) | $\theta_A^{\tau hu} = (\mu_A^{\tau hu}, \sigma_A^{\tau hu})$ | $\mu_A^{\tau hu} \sim$ Gaussian( 0, 100 ) <br> $\sigma_A^{\tau hu} \sim$ Gamma( 1, 0.001 ) |
| spatial bias, $B^{\tau shu}$ | $B^{\tau shu} \sim$ Gaussian($\mu_B^{\tau hu}, \sigma_B^{\tau hu}$) | $\theta_B^{\tau hu} = (\mu_B^{\tau hu}, \sigma_B^{\tau hu})$ | $\mu_B^{\tau hu} \sim$ Gaussian( 0, 100 ) <br> $\sigma_B^{\tau hu} \sim$ Gamma( 1, 0.001 ) |
| decision noise, $\sigma^{\tau shu}$ | $\sigma^{\tau shu} \sim$ Gamma($k_\sigma^{\tau hu}, \lambda_\sigma^{\tau hu}$) | $\theta_\sigma^{\tau hu} = (k_\sigma^{\tau hu}, \lambda_\sigma^{\tau hu})$ | $k_\sigma^{\tau hu} \sim$ Exp( 0.1 ) <br> $\lambda_\sigma^{\tau hu} \sim$ Exp( 10 ) |

DOI: https://doi.org/10.7554/eLife.27430.011

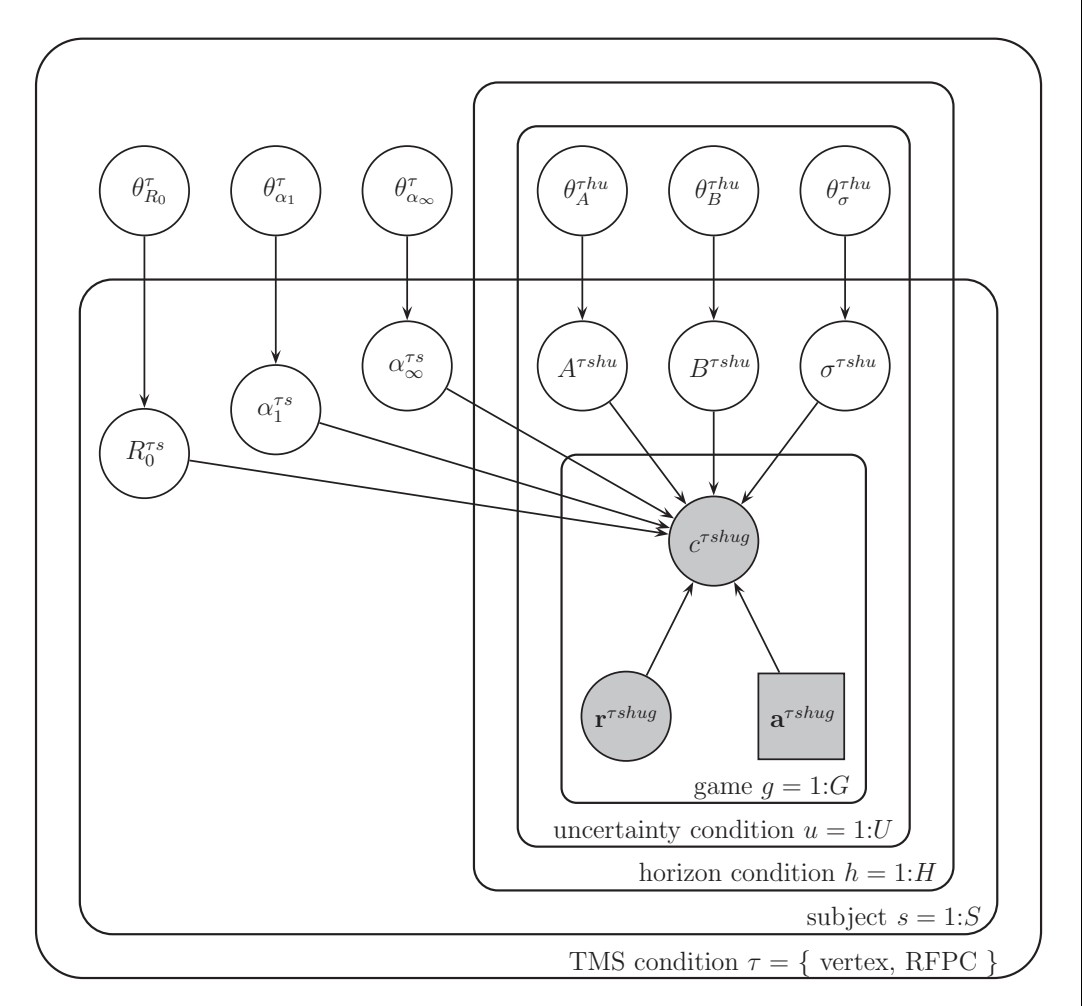

**Figure 8.** Graphical representation of the model. Each variable is represented by a node, with edges denoting the dependence between variables. Shaded nodes correspond to observed variables, that is, the free choices $c^{\tau shug}$, forced-trial rewards, $\mathbf{r}^{\tau shug}$ and forced-trial choices $\mathbf{a}^{\tau shug}$. Unshaded nodes correspond to unobserved variables whose values are inferred by the model.

DOI: https://doi.org/10.7554/eLife.27430.010

$$p(\text{choose right}) = \frac{1}{1 + \exp\left(\frac{\Delta R + A\Delta I + B}{\sigma}\right)} \tag{12}$$

where $\Delta R$ ( $= R_t^{left} - R_t^{right}$ ) is the difference in expected reward between left and right options and $\Delta I$ is the difference in information between left and right options (which we define as +1 when left is more informative, $-1$ when right is more informative, and 0 when both options convey equal information in the (**Auer et al., 2002**; **Auer et al., 2002**) condition). The three free parameters of the decision process are: the information bonus, $A$, the spatial bias, $B$, and the decision noise $\sigma$. We assume that these three decision parameters can take on different values in the different horizon and uncertainty conditions (with the proviso that $A$ is undefined in the (**Auer et al., 2002**; **Auer et al., 2002**) information condition since $\Delta I = 0$). Thus the decision component of the model has 10 free parameters ($A$ in the two horizon conditions, and $B$ and $\sigma$ in the 4 horizon x uncertainty conditions). Directed exploration is then quantified as the change in information bonus with horizon, while random exploration is quantified as the change in decision noise with horizon.

## Model fitting

### Hierarchical bayesian model

Between the learning and decision components of the model, each subject's behavior is described by 13 free parameters, all of which are allowed to vary between TMS conditions. These parameters are: the initial mean, $R_0$, the initial learning rate, $\alpha_1$, the asymptotic learning rate, $\alpha_\infty$, the information bonus, $A$, in both horizon conditions, the spatial bias, $B$, in the four horizon x uncertainty conditions, and the decision noise, $\sigma$, in the four horizon x uncertainty conditions (*Table 2*, *Figure 8*).

Each of the free parameters is fit to the behavior of each subject using a hierarchical Bayesian approach (*Lee and Wagenmakers, 2014*). In this approach to model fitting, each parameter for each subject is assumed to be sampled from a group-level prior distribution whose parameters, the so-called 'hyperparameters', are estimated using a Markov Chain Monte Carlo (MCMC) sampling procedure. The hyper-parameters themselves are assumed to be sampled from 'hyperprior' distributions whose parameters are defined such that these hyperpriors are broad. For notational convenience, we refer to the hyperparameters that define the prior for variable X as $\theta^X$. In addition we use subscripts to refer to the dependence of both parameters and hyperparameters on TMS stimulation condition, $\tau$, horizon condition, $h$, uncertainty condition, $u$, subject, $s$, and game, $g$.

The particular priors and hyperpriors for each parameter are shown in *Table 2*. For example, we assume that the prior mean, $R_0^{\tau s}$, for each stimulation condition $\tau$ and horizon condition $h$, is sampled from a Gaussian prior with mean $\mu_{R_0}^\tau$ and standard deviation $\sigma_{R_0}^\tau$. These prior parameters are sampled in turn from their respective hyperpriors: $\mu_{R_0}^\tau$, from a Gaussian distribution with mean 50 and standard deviation 14, $\sigma_{R_0}^\tau$ from a Gamma distribution with shape parameter 1 and rate parameter 0.001.

### Model fitting using MCMC

The model was fit to the data using Markov Chain Monte Carlo approach implemented in the JAGS package (*Plummer, 2003*) via the MATJAGS interface (psiexp.ss.uci.edu/research/programs_data/jags/). This package approximates the posterior distribution over model parameters by generating samples from this posterior distribution given the observed behavioral data.

In particular we used 4 independent Markov chains to generate 4000 samples from the posterior distribution over parameters (1000 samples per chain). Each chain had a burn in period of 500 samples, which were discarded to reduce the effects of initial conditions, and posterior samples were acquired at a thin rate of 1. Convergence of the Markov chains was confirmed post hoc by eye. Code and data to replicate our analysis and reproduce our Figures is provided as part of the Supplementary Materials.

## Additional information

### Funding

No external funding was received for this work.

### Author contributions

Wojciech K Zajkowski, Conceptualization, Formal analysis, Investigation, Visualization, Methodology, Writing—original draft, Project administration, Writing—review and editing; Malgorzata Kossut, Resources, Supervision, Project administration; Robert C Wilson, Conceptualization, Software, Formal analysis, Supervision, Visualization, Writing—original draft, Writing—review and editing

### Author ORCIDs

Robert C Wilson https://orcid.org/0000-0002-2963-2971

### Ethics

Human subjects: All participants were informed about potential risks connected to TMS and signed a written consent. The study was approved by University of Social Sciences and Humanities ethics committee.

Decision letter and Author response
Decision letter https://doi.org/10.7554/eLife.27430.020
Author response https://doi.org/10.7554/eLife.27430.021

# Additional files

## Supplementary files
• Transparent reporting form
DOI: https://doi.org/10.7554/eLife.27430.012

## Major datasets
The following dataset was generated:

| Author(s) | Year | Dataset title | Dataset URL | Database, license, and accessibility information |
|---|---|---|---|---|
| Zajkowski, W, Kossut, M, Wilson, RC | 2017 | A causal role for right frontopolar cortex in directed, but not random, exploration | https://dataverse.harvard.edu/dataset.xhtml?persistentId=doi:10.7910/DVN/CZT6EE | Publicly accessible via the Harvard Dataverse website (https://dx.doi.org/10.7910/DVN/CZT6EE) |

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
