## [Decision Letter]

[Editors’ note: a previous version of this study was rejected after peer review, but the authors submitted for reconsideration. The first decision letter after peer review is shown below.]

Thank you for submitting your work entitled "A causal role for right frontopolar cortex in directed, but not random, exploration" for consideration by *eLife*. Your article has been reviewed by two peer reviewers, and the evaluation has been overseen by a Reviewing Editor and a Senior Editor. The reviewers have opted to remain anonymous.

Our decision has been reached after consultation between the reviewers. Based on these discussions and the individual reviews below, we regret to inform you that your work in the current state will not be considered further for publication in *eLife*.

All involved found the work to have great merit and contributes to the literature on RLPFC and exploration. In our view this is perhaps the clearest demonstration to date that the RLPFC is involved in directed, uncertainty-guided exploration, in that it is the first to imply causality. However, given the state of the literature with other studies (cited in your manuscript) that show RLPFC activation during exploration, that it codes for uncertainty and/or the value of alternative actions, together with an existing TDCS study manipulating it and affecting exploration (albeit not in a way that clearly implicates uncertainty), we felt that the bar for establishing causality in your study needs to be quite high. The reviewers agreed that given the small sample and somewhat marginal statistics, it would be more reassuring if the results held up in a larger N study (or a separate independent replication). Moreover, while the findings here are compelling (e.g. the selectivity to horizon 6 directed exploration), they would be more so especially if you had a control site of stimulation (e.g. DLPFC or IFG) to establish specificity of the RLPFC site. (One of the reviewers noted in the consultation session that RLFPC stimulation may cause discomfort relative to vertex stimulation, which could differentially impact conditions that may require differences in effort).

Reviewer 1 also had concerns regarding potential power differences to detect effects in directed vs. random exploration.

If you feel strongly that you can address these concerns we could consider a resubmission. But because the nature of the concerns requires new data collection, and it is unclear whether the results of new studies will provide more clarity, we are rejecting the paper as it stands. We would understand if you chose to submit this study as it is elsewhere.

*Reviewer #1:*

This manuscript reports the results of a TMS study in which participants are stimulated with theta-burst TMS while participating in a one armed bandit gambling task aimed at distinguishing directed from random exploration. The authors hypothesize that frontopolar cortex is involved in directed but not random exploration. Using both model-based and model-free analyses the authors report that frontopolar cortex inhibition impacts on directed but not random exploration, allowing the authors to conclude that this structure plays a specific role in directed exploration.

Overall, the study is an interesting one that identifies a potentially important finding. The notion that frontopolar cortex is especially involved in directed exploration is highly plausible, and the results do indeed provide some indication of this possibility. However, I do have several major concerns which I detail below.

1) One major concern is the possibility that there is a substantial difference in power to detect the effects of TMS on the two forms of exploration due to perhaps a big difference in the number of behavioral choice that index these two forms of exploration on the first trial of each block. While directed exploration in the vertex treatment in the [15] condition perhaps occurs frequently (the number of trials in which this behavior is found are not reported, but I am inferring this from the high probability of directed exploration reported in the horizon 6 condition), it seems natural to expect that there would be much fewer instances of "random" exploration as defined by choice of the lower valued option in the [24] condition – this appears to be reflected in the much lower reported probabilities of random exploration in that condition. If there are many fewer trials of random exploration in the first place this ought to make an effect of random exploration following TMS stimulation much harder to detect. Therefore, one trivial account for the authors' double dissociation is that it occurs as a result of a difference in the experimental power to detect these two effects in the paradigm. The claim the authors have about an effect of TMS on direct exploration per se seems well supported in my opinion, but the claim for the specificity of the effect to random exploration seems a lot weaker.

2) Another concern is that for a behavioral and TMS study the use of such a small sample size of only 15 participants seems hard to justify, especially given that the authors are reporting effects that are just barely reaching significance at p<0.05. Given the concerns raised above about power to detect effects on random exploration, and given that there are a very small number of trials per subject enabling the authors to test their claimed effects (as they are throwing away most of the trials per block and focusing only on the first), suggest that it would not be unreasonable to expect the authors to obtain a larger sample size.

3) A more generic concern with TMS over frontopolar cortex is that it is unclear with this stimulation protocol how diffuse the effect of the TMS stimulation has been, and to what extent the stimulation protocol has also impacted adjacent regions of frontal cortex. This is an inherent limitation of this technique of course, but there are ways to ameliorate concerns in this regard such as by measuring effects of the stimulation protocol with fMRI. The authors could discuss this limitation and ideally bolster their claims about the degree to which these effects can be specifically attributed to effects of stimulation on frontopolar cortex per se.

4) Could the apparent effect on directed exploration be driven by other more prosaic possibilities such as an impairment in the ability to flexibly change task set (e.g. from a short to long horizon) across blocks or alterations in the capacity to attend to the task cues indicating the horizon length or even the capacity to incorporate knowledge of task instructions could be impacted instead of directed exploration per se.

5) Can the authors discriminate between different ways in which directed exploration could be implemented computationally on this task? For instance one could imagine a Bayesian implementation in which a representation of uncertainty over the options is computed and used to direct exploration toward the more uncertain options, or else one could simply use a heuristic strategy of just counting the number of samples of each option to try to ensure each option has been sampled an equivalent number of times.

6) Although the authors cite Wilson et al. (2014) to describe their modeling strategies, it would be important to reproduce details of exactly how they implemented the model fitting etc. in the current paradigm, as these analyses are central to the current paper and the reader shouldn't be required to go searching for another paper to understand precisely what was done.

7) I wonder whether more use can be made of the subsequent trials in each block. It seems a shame to throw these trials away, even if the utility of the trials for distinguishing these constructs drops off over repeated trials within a block it seems plausible to me that the 2nd and 3rd trials at the very least would contain useful information.

*Reviewer #2:*

Zajkowski and colleagues present a study showing that continuous theta burst stimulation to right frontopolar cortex, but not the vertex, selectively reduces directed exploration, but not random exploration. I commend the authors for their experimental approach, combining a carefully designed experimental paradigm and computational modeling of behavior with a transient causal manipulation, such as cTBS. While the results look straightforward, and I do believe they represent an advance on current knowledge in the field, I do not think they represent such a significant advance to merit publication in *eLife* (or a similar high impact journal of broad interest), but would be appropriate for a more specialized journal in the field. Rather than advancing thinking on this topic in some new way, developing a new methodology, or resolving a debate, I believe the results essentially confirm what could be inferred to be likely from the existing fMRI (Daw et al., 2006; Badre et al., 2012) and stimulation (TDCS) literature (Beharelle et al., 2015) on the RFPC and exploration/exploitation. Furthermore, the experimental paradigm and modeling results have been published (Wilson, et al., 2014). I do not mean to discourage the authors, who I think have conducted a genuinely interesting study by combining approaches in an unusual way, and confirming their main hypothesis. I simply do not believe the paper is best suited for a journal of the caliber of *eLife*, but I of course leave this up to the editor's discretion. I have added a few comments below that I hope will be helpful to the authors.

In the Introduction random exploration is framed as simply increasing decision noise, and directed exploration as information seeking. But is that really the critical distinction? In the real world random exploration is likely to occur when the environmental statistics have changed very rapidly and/or the animal has inferred (for whatever reason) their prior causal model (or even set of models) is (are) no longer tenable. In these circumstances their exploratory behavior is likely still characterized as information seeking, even if it manifests formally as an increase in decision noise. It seems to me, therefore, that the key distinction between directed and undirected exploration is that animals no longer know which options to explore. Can the authors clarify their view, and perhaps modify the Introduction and/or Discussion as needed?

What were the instructions to participants? Do they necessarily understand that the bandit means are constant and independent? Is there any evidence they weight more recent past samples more strongly than more distant samples? Would this change the estimates of the means in any meaningful way?

Given the demonstrated effects of cTBS on the hemodynamic signal measured in control networks (Gratton et al., 2013), how specific is the effect of stimulation to RFPC? To address this question, I would have liked to see the investigators target another frontal comparison brain region, in addition or instead of the vertex.

[Editors’ note: what now follows is the decision letter after the authors submitted for further consideration.]

Thank you for resubmitting your work entitled "A causal role for right frontopolar cortex in directed, but not random, exploration" for further consideration at *eLife*. Your revised article has been favorably evaluated by Sabine Kastner as Senior Editor, Michael Frank as Reviewing editor and two reviewers.

The manuscript has been improved, especially given the doubled sample size, and the model-based and model-free analyses are sophisticated, comprehensive, and generally compelling. However, there are some remaining issues that need to be addressed before acceptance, as outlined below:

1) Why do the authors binarize relative information such that it is coded as +1 when the left gamble is more informative and -1 when the right gamble is? Based on Badre, Doll, Frank, et al. I would have thought that the estimated relative uncertainty between options would be more appropriate to quantitatively test the impact of stimulation on directed exploration. Or is variance in this quantity negligible across the critical choices in this task? Related to this question, is this quantity matched across conditions and do all subjects see identical or different schedules?

2) Although I am not requesting the authors conduct another experiment, a second stimulation site within prefrontal cortex would make for an important comparison for future studies. My suggestion is in part due to the quite severe discomfort frequently caused by TMS stimulation to FPC and neighboring regions due to the underlying facial musculature, as compared to say the vertex. Any differences between stimulation sites could in theory be due to differences in discomfort or subsequent distraction produced by the stimulation sites. Here, this difference could conceivably interact with the comparison between horizon 6 and horizon 1 in the unequal condition if this horizon 6 condition is in fact more cognitively demanding. Note this is not a concern in the cited tDCS study by Raja Beharelle et al. because tDCS does not stimulate the facial muscles and because the excitation and inhibition respectively following anodal and cathodal tDCS provides for an internal control. Can the authors provide some evidence that horizon 6 in the unequal condition is not the most cognitively demanding for their subjects, for instance by analysing RTs? Are there existing data that address this concern by comparing stimulation of FPC and other PFC regions using cTBS?

3) The trend of an effect of RFPC stimulation on the information bonus for horizon 1, although smaller than that of horizon 6, seems problematic for an interpretation purely based on directed exploration, since there is no opportunity to exploit the newly acquired information for horizon 1. The authors suggest subjects may become less information-seeking in both conditions (consistent with risk or ambiguity aversion in horizon 1 and reduced directed exploration in horizon 6), but this begs the question of what process or mechanism underlies this decrease in both horizons. Given the broader literature on the role of FPC, one interpretation would be that stimulation has disrupted the FPC's ability to faithfully encode the parameters of a "pending" option that they may choose in the future (e.g. Koechlin and Hyafil, Science, 2007) – in this task this could be seen as the option that has not been selected as frequently or attended to recently during forced choices. However I am sure there are other plausible interpretations. How do the authors interpret this effect across horizons in the unequal condition with respect to the broader literature on FPC?

---

## [Author Response]

[Editors’ note: the author responses to the first round of peer review follow.]

Reviewer #1:[…] 1) One major concern is the possibility that there is a substantial difference in power to detect the effects of TMS on the two forms of exploration due to perhaps a big difference in the number of behavioral choice that index these two forms of exploration on the first trial of each block. While directed exploration in the vertex treatment in the [1 3] condition perhaps occurs frequently (the number of trials in which this behavior is found are not reported, but I am inferring this from the high probability of directed exploration reported in the horizon 6 condition), it seems natural to expect that there would be much fewer instances of "random" exploration as defined by choice of the lower valued option in the [2 2] condition – this appears to be reflected in the much lower reported probabilities of random exploration in that condition. If there are many fewer trials of random exploration in the first place this ought to make an effect of random exploration following TMS stimulation much harder to detect. Therefore, one trivial account for the authors' double dissociation is that it occurs as a result of a difference in the experimental power to detect these two effects in the paradigm. The claim the authors have about an effect of TMS on direct exploration per se seems well supported in my opinion, but the claim for the specificity of the effect to random exploration seems a lot weaker.

This is an important point. Put simply, do we find no effect on random exploration because our experiment is underpowered to detect effects on random exploration? We believe that we do have sufficient power to detect an effect on random exploration (if it were there) and we try to show this using both a model-free and model-based approach.

For the model-free approach we consider the size of the horizon effect for directed and random exploration in the control condition. This horizon effect is essentially the effect we are trying to remove with TMS and the idea is that, if the horizon effect size is smaller for random than directed exploration, there would be a difference in power to detect changes to the horizon effect. Fortunately the horizon effects are of equal size in this study (in the vertex condition Cohen’s d for directed = 0.71; for random = 0.68). These numbers are largely in line with pure behavioral subjects (the 60 undergraduates from Somerville et al. 2016) where we find d = 0.75 for directed and, a slightly larger, d = 1.18 for random. Thus, if TMS were to reduce the horizon effect by 50% we would have essentially equal power to detect both effects (note we have the same number of trials in the [2 2] condition, for measuring p(low mean) and random exploration, and [1 3] condition, for measuring *p*(high info) and directed exploration).

For the model-based approach, we can fit the decision noise in the [1 3] uncertainty condition in addition to the [2 2] condition. This gives us an independent estimate of decision noise and gives us another chance to see an effect of TMS on random exploration. In addition, in our new model-based analysis, we use hierarchical Bayesian model fitting to compute posterior distributions over all model parameters given the data (see reviewer #1 response #6 for more details on this model). As shown by the posterior distributions (Figure 4, main text) we see no effect of TMS on decision noise in *any* of the four uncertainty x horizon conditions, but we do see an effect on information bonus in horizon 6.

2) Another concern is that for a behavioral and TMS study the use of such a small sample size of only 15 participants seems hard to justify, especially given that the authors are reporting effects that are just barely reaching significance at p<0.05. Given the concerns raised above about power to detect effects on random exploration, and given that there are a very small number of trials per subject enabling the authors to test their claimed effects (as they are throwing away most of the trials per block and focusing only on the first), suggest that it would not be unreasonable to expect the authors to obtain a larger sample size.

We agree that N = 15 was not ideal. We have now run an additional 16 subjects and our results hold (see Author response image 1).

**Author response image 1. respfig1:** No difference in effects between original and replication experiments. In each panel we plot the model-free measures of directed and random exploration and how they change between stimulation conditions. For example, in Panel A, we plot *p*(high info) in horizon 1 for vertex stimulation (x-axis) and RFPC stimulation (y-axis). Each point in this plot is a single subject and the diagonal line represents equality. Participants below the diagonal line have a smaller value of *p*(high info) in the RFPC stimulation condition. From this we can clearly see that there is no effect of RFPC stimulation on directed exploration in horizon 1 (panel A), or random exploration in either horizon (B, D). However, there is a strong effect of RFPC stimulation on directed exploration in horizon 6 with the majority of points lying below the diagonal (C). Moreover, both the original and replication datasets point to the same conclusions in all four panels.

In addition, we have included two new analyses: a model-based Bayesian analysis (results of which are shown in Figure 4), as well as a model-free analysis of later trials. Both of these analyses point to the same conclusion – inhibition of RFPC leads to selective inhibition of directed exploration in horizon 6.

The model-free analysis of later trials is presented in the main paper in Figure 6 in its own section. In this analysis we compute *p*(high info) and *p*(low mean) for all trials in the horizon 6 game to see whether behavior on the later trials is affected by stimulation of frontal pole. For directed exploration we find some evidence that the reduction in *p*(high info) on the first trial continues into the second (post hoc, one-sided t-test on the second trial, t(24) = 1.61; p = 0.06), Figure 6 panels A and C. While this is a marginal result, it is consistent with our hypothesis and provides more support for frontal pole playing a role in directed exploration. For random exploration we find no effect of RFPC stimulation on any trial. This is consistent with the idea that frontal pole is not involved in random exploration.

For completeness we reproduce the particular section of text here:

“The effect of RFPC stimulation on later trials

Our analyses so far have focused on just the first free choice and have ignored the remaining five choices in the horizon 6 games. […] Thus, the analysis of later trials provides additional, albeit modest, support for the idea that RFPC stimulation selectively disrupts directed but not random exploration at long horizons.”

3) A more generic concern with TMS over frontopolar cortex is that it is unclear with this stimulation protocol how diffuse the effect of the TMS stimulation has been, and to what extent the stimulation protocol has also impacted adjacent regions of frontal cortex. This is an inherent limitation of this technique of course, but there are ways to ameliorate concerns in this regard such as by measuring effects of the stimulation protocol with fMRI. The authors could discuss this limitation and ideally bolster their claims about the degree to which these effects can be specifically attributed to effects of stimulation on frontopolar cortex per se.

We agree that this is an important point and would be an important follow-up study. We have added the following to the Discussion to address this point:

“While the present study does allow us to conclude that directed and random exploration rely on different neural systems, the limited spatial specificity of TMS limits our ability to say exactly what those systems are. […] Future work combining cTBS with neuroimaging will be necessary to shed light on these issues.”

4) Could the apparent effect on directed exploration be driven by other more prosaic possibilities such as an impairment in the ability to flexibly change task set (e.g. from a short to long horizon) across blocks or alterations in the capacity to attend to the task cues indicating the horizon length or even the capacity to incorporate knowledge of task instructions could be impacted instead of directed exploration per se.

This is an interesting idea that we believe we can rule out. To paraphrase, the idea is that RFPC stimulation inhibits the ability to adapt to horizon in general (e.g. by causing subjects to ignore relevant task cues) rather than causing a specific deficit in directed exploration. Such a general deficit would predict that the horizon effect on random exploration would also be abolished with RFPC stimulation and this is something we do not see at all in three separate analyses.

First, in the model-free analysis (Figure 3) we see that *p*(low mean) increases with horizon even in the RFPC condition and that RFPC stimulation has no effect on this measure of random exploration.

Second, this model-free result also holds for the later trials in which we see no stimulation based change in *p*(low mean) over the course of horizon 6 games (Figure 6). Of course, these later trial results are subject to the reward information confound and so should not be overinterpreted, but they do at least point to the same conclusion that RFPC stimulation does not change the horizon dependence of random exploration.

Third, our model-based analysis points to the same conclusion that there is no change in decision noise with stimulation condition (Figure 4).

5) Can the authors discriminate between different ways in which directed exploration could be implemented computationally on this task? For instance one could imagine a Bayesian implementation in which a representation of uncertainty over the options is computed and used to direct exploration toward the more uncertain options, or else one could simply use a heuristic strategy of just counting the number of samples of each option to try to ensure each option has been sampled an equivalent number of times.

Unfortunately the vanilla Horizon Task used here is not well suited to addressing this question. The reason is that uncertainty on the first free choice is not parametrically modulated – there either is a difference in uncertainty (in the [1 3] condition) or else there is no difference in uncertainty (in the [2 2] condition). While one could try to look at this with a model-based analysis of the later trials, such an analysis is deeply affected by the reward-information confound which makes interpreting results of such an analysis difficult.

In an on-going set of experiments, we have performed a (purely behavioral) version of the task with parametric modulation of uncertainty. This reveals that the information bonus does appear to scale with uncertainty in a more Bayesian manner, more analysis needs to be done to be sure and the result requires internal replication (much easier with pure behavior than TMS!) before we publish.

6) Although the authors cite Wilson et al. (2014) to describe their modeling strategies, it would be important to reproduce details of exactly how they implemented the model fitting etc. in the current paradigm, as these analyses are central to the current paper and the reader shouldn't be required to go searching for another paper to understand precisely what was done.

This is a fair point and we have now included much more detail on the model. In addition we have expanded the model to include a learning component and fit the model in a different (and more rigorous) hierarchical Bayesian manner. We describe the model at two different points in the text and provide code to implement the model in the Supplementary Material. In the Results section, we highlight the salient points to try to convey the main intuition in the subsection “RFPC stimulation selectively inhibits directed exploration on the first free-choice”. In the Materials and methods section, we go into all the gory details. As this text is extensive, we do not quote it here.

7) I wonder whether more use can be made of the subsequent trials in each block. It seems a shame to throw these trials away, even if the utility of the trials for distinguishing these constructs drops off over repeated trials within a block it seems plausible to me that the 2nd and 3rd trials at the very least would contain useful information.

I wonder this too and have been for quite a while! In trying to model the later trials, it quickly becomes apparent that the reward-information confound is very real and introduces very strong correlations between the fitted parameter values that makes interpretation of the results essentially impossible.

Despite this difficulty in interpreting the model-based parameters, the model-free statistics (while still being confounded) are at least more straightforward. As mentioned above (response #2), we include this model-free analysis of later trials in a separate section of the Results, along with appropriate health warnings about the reward information confound.

Reviewer #2:Zajkowski and colleagues present a study showing that continuous theta burst stimulation to right frontopolar cortex, but not the vertex, selectively reduces directed exploration, but not random exploration. I commend the authors for their experimental approach, combining a carefully designed experimental paradigm and computational modeling of behavior with a transient causal manipulation, such as cTBS.

We thank the reviewer for the positive comments and helpful feedback. We hope this revision will change your mind about the “importance” of the findings, but regardless of whether the paper is accepted to *eLife*, your comments have greatly improved the paper!

While the results look straightforward, and I do believe they represent an advance on current knowledge in the field, I do not think they represent such a significant advance to merit publication in eLife (or a similar high impact journal of broad interest), but would be appropriate for a more specialized journal in the field. Rather than advancing thinking on this topic in some new way, developing a new methodology, or resolving a debate, I believe the results essentially confirm what could be inferred to be likely from the existing fMRI (Daw et al., 2006; Badre et al., 2012) and stimulation (TDCS) literature (Beharelle et al., 2015) on the RFPC and exploration/exploitation. Furthermore, the experimental paradigm and modeling results have been published (Wilson, et al., 2014). I do not mean to discourage the authors, who I think have conducted a genuinely interesting study by combining approaches in an unusual way, and confirming their main hypothesis. I simply do not believe the paper is best suited for a journal of the caliber of eLife, but I of course leave this up to the editor's discretion. I have added a few comments below that I hope will be helpful to the authors.

While we acknowledge that such judgments of “importance” are often a matter of taste and perspective (all things look big when viewed up close!), we respectfully disagree with this point and believe our study represents a major update to current thinking. In particular, by showing that RFPC stimulation selectively inhibits directed exploration we show that “exploration” is not a *unitary* process, it is a *dual* process in which directed and random exploration rely on (at least partially) dissociable neural systems.

That exploration is a dual process is absolutely not something one would have concluded from previous work. For example, Daw and Badre see similar activations despite defining exploration in very different ways (choosing low value option for Daw and (loosely) choosing high information options for Badre). The reason the activations are similar is that both tasks have a reward-information confound and after making just a few free choices, the high information options *are* the low value options. This means that every single exploration-related activation in those studies now has a big question mark on it – is it an activation related to directed exploration, random exploration or both? The same can be said of the Beharelle finding, which is beautiful in how it shows opposite effects for anodal and cathodal stimulation, but which cannot dissociate directed and random exploration because of the nature of the behavioral task. To be clear, we do not mean to attack previous work here – these are all incredibly important studies. However, our findings do open them up to reinterpretation.

We have tried to emphasize this dual-process interpretation in the Discussion:

“In this work we used continuous theta-burst transcranial magnetic stimulation (cTBS) to investigate whether right frontopolar cortex (RFPC) is causally involved in directed and random exploration. […] This is consistent with the idea that the levels of directed and random exploration are set by the strength of an exploratory drive that varies as an individual difference between people.”

In the Introduction random exploration is framed as simply increasing decision noise, and directed exploration as information seeking. But is that really the critical distinction? In the real world random exploration is likely to occur when the environmental statistics have changed very rapidly and/or the animal has inferred (for whatever reason) their prior causal model (or even set of models) is (are) no longer tenable. In these circumstances their exploratory behavior is likely still characterized as information seeking, even if it manifests formally as an increase in decision noise. It seems to me, therefore, that the key distinction between directed and undirected exploration is that animals no longer know which options to explore. Can the authors clarify their view, and perhaps modify the Introduction and/or Discussion as needed?

This is a really interesting idea and one that would be worth investigating in its own right. We have added a few sentences to the Discussion suggesting that random exploration may be a “model-free” method of exploration that works especially well when the model is unknown.

“With the above caveats that our results may not be entirely due to disruption of frontal pole, the interpretation that RFPC plays a role in directed, but not random, exploration is consistent with a number of previous findings. […] Indeed, the ability to explore effectively in a model-free manner, may be an important function of random exploration as it allows us to explore even when our model of the world is wrong.”

What were the instructions to participants? Do they necessarily understand that the bandit means are constant and independent?

The instructions were a direct Polish translation of the original instructions used by Wilson et al. (2014). These instructions clearly state that the average reward from each bandit is constant in each game and that the variability is constant over the entire game. For reference see the supplementary material of the original paper. If you feel it would be important for this paper, we would be happy to include them as Supplementary Material.

Is there any evidence they weight more recent past samples more strongly than more distant samples? Would this change the estimates of the means in any meaningful way?

This is a great question and one that has pushed us to update the model. In particular, we have now modeled the learning process (i.e. the process by which participants infer the mean of each option from the forced trials) using a Kalman filter. This model assumes that participants learn the mean reward for each option using a delta-rule update equation

*R^i^_t+1_ = R^i^_t_ + α^i^_t_ (r_t_ – R^i^_t_)* (*)

Where *α^i^_t_* is the time varying learning rate. The time dependence of the learning rate is determined by the Kalman filter equations (see Materials and methods for full description of the model) and can be parameterized by two parameters: the initial learning rate *α_0_* and the asymptotic learning rate *α_inf_*. Crucially, equation (*) allows for potentially uneven weighting of the reward depending on the values of *α_0_* and *α_inf_*. Our previous model, with equal weighting given to all points, corresponds to the case of *α_0_*= 1, *α_inf_* = 0. Models with *α_0_*< 1 and *α_inf_* > 0 have a recency bias, weighting more recent rewards more strongly.

The posterior distributions over the group average values of *α_0_*and *α_inf_* are shown in Figure 3 in the main paper. In particular *α_0_*~ 0.6 and *α_inf_* ~ 0.45, suggesting quite a pronounced recency effect. Importantly, however, neither of these parameters changes between stimulation conditions, and including this learning term in the model does not change the effect of TMS on directed exploration (information bonus in horizon 6).

Given the demonstrated effects of cTBS on the hemodynamic signal measured in control networks (Gratton et al., 2013), how specific is the effect of stimulation to RFPC? To address this question, I would have liked to see the investigators target another frontal comparison brain region, in addition or instead of the vertex.

We agree that our inability to nail down the specificity of the effect is an important limitation of this work. Unfortunately we currently lack the resources to run a study looking at stimulation of other areas and have instead focused our efforts on increasing the sample size of the current study. Likewise, combining TMS with fMRI will be important in future work to more precisely characterize the effects of the perturbation. We have acknowledged both of these limitations in the Discussion as follows:

“While the present study does allow us to conclude that directed and random exploration rely on different neural systems, the limited spatial specificity of TMS limits our ability to say exactly what those systems are. […] Future work combining cTBS with neuroimaging will be necessary to shed light on these issues.”

[Editors' note: the author responses to the re-review follow.]

1) Why do the authors binarize relative information such that it is coded as +1 when the left gamble is more informative and -1 when the right gamble is? Based on Badre, Doll, Frank, et al. I would have thought that the estimated relative uncertainty between options would be more appropriate to quantitatively test the impact of stimulation on directed exploration. Or is variance in this quantity negligible across the critical choices in this task? Related to this question, is this quantity matched across conditions and do all subjects see identical or different schedules?

There are a few thoughts behind binarizing information. First, binary information matches the task design in which there is only one unequal information condition and no gradations in uncertainty from a normative perspective. Related to this, and as the reviewer rightly intuits, the single unequal uncertainty condition means that the variance in relative uncertainty between options is relatively small meaning that there is very little difference between the binarized vs continuous definition of information. Because of this we have decided to stick with the binarized version in the paper so as to avoid over interpreting the data.

More generally, the parametric effect of uncertainty in this task is a key question and is something we are looking at behaviorally in ongoing experiments with different numbers of forced trials. Such explicit manipulation of information leads to much more variance in the uncertainties allowing us to compute parametric effects of uncertainty with more confidence. In brief, these results do suggest a linear effect of uncertainty as seen in previous work.

Of course, it is possible to fit the continuous model to the data in this paper and when we do so we come to the same conclusions as the binarized model – a selective effect of RFPC stimulation on directed exploration (see Author response image 2 and Author response image 3).

**Author response image 2. respfig2:** Effect of TMS on information bonus in model with bonus proportional to uncertainty.

**Author response image 3. respfig3:** Effect of TMS on decision noise in model in which bonus is a linear function of uncertainty.

Finally, as to the question of whether participants receive exactly the same schedule of trials or not, unfortunately this was not perfectly controlled in either direction. The first 16 subjects (the initial group) were run with the same random seed while the remaining subjects (the replication group) were run with unique random seeds. Given the results replicate between groups we do not think this is a major issue although we now include the following text in the Materials and methods section:

“Finally, the random seeds were not perfectly controlled between subjects. […] Despite this limitation we saw no evidence of different behavior across the two groups.”

2) Although I am not requesting the authors conduct another experiment, a second stimulation site within prefrontal cortex would make for an important comparison for future studies. My suggestion is in part due to the quite severe discomfort frequently caused by TMS stimulation to FPC and neighboring regions due to the underlying facial musculature, as compared to say the vertex. Any differences between stimulation sites could in theory be due to differences in discomfort or subsequent distraction produced by the stimulation sites. Here, this difference could conceivably interact with the comparison between horizon 6 and horizon 1 in the unequal condition if this horizon 6 condition is in fact more cognitively demanding. Note this is not a concern in the cited tDCS study by Raja Beharelle et al. because tDCS does not stimulate the facial muscles and because the excitation and inhibition respectively following anodal and cathodal tDCS provides for an internal control. Can the authors provide some evidence that horizon 6 in the unequal condition is not the most cognitively demanding for their subjects, for instance by analysing RTs?

We agree that other types and locations of stimulation will be an important avenue for future work and is something that I (RCW) am planning once TMS becomes available at UA.

The point about pain is also important. As we understand it, the idea is that RFPC stimulation can be painful. Pain is distracting which leads to worse performance, especially when a task is cognitively demanding. Thus if directed exploration in horizon 6 is the most cognitively demanding component of the task, then distraction from pain could cause the effect.

While we cannot rule this interpretation out entirely, two results suggest that simple distraction is likely not to blame.

First, one prediction of the distraction hypothesis is that people should perform worse overall when distracted by pain. In the model-free analysis this should show up as increased *p*(low mean) with stimulation of frontal pole. In the model-based analysis, distraction should manifest as increased decision noise in both [1 3] and [2 2] conditions. In both analyses we see no effect of RFPC stimulation (Figure 3 and Figure 4). This effectively puts an *upper bound* on how distracting the pain could be – the distraction effect must be small enough to cause no change in the ability to pick out the high reward option.

Of course, the above analysis says nothing about the *lower bound* and it could still be the case that, while the pain is not distracting enough to affect computing the mean reward, it *is* distracting enough to affect the computations of the information bonus. This could be the case if computing the information bonus were harder than computing the mean. Evidence for this increased computational load could come from reaction times. Specifically if computing the bonus is difficult, then RTs should be longer in the [1 3] condition in horizon 6 than in horizon 1. As shown in Author response image 4 this is not the case and there is no effect of horizon on RT for the first free choice (F = 1.32, p = 0.26). Thus computing the information bonus is not a time consuming process, suggesting it is not any more taxing than computing the difference in means between options.

**Author response image 4. respfig4:** 

Together with the null effect on *p*(low mean) we believe that these results provide good evidence that our effects are driven by neural changes (presumably in RFPC – although this is impossible to verify without neuroimaging) not as a response to pain.

Are there existing data that address this concern by comparing stimulation of FPC and other PFC regions using cTBS?

A Google Scholar search for “cTBS frontal pole” found only one paper that reported pain measures. None that we could find directly compared pain from stimulation to FPC and other areas of PFC.

Hanlon, C. A., Dowdle, L. T., Correia, B., Mithoefer, O., Kearney-Ramos, T., Lench, D.,[…] and George, M. S. (2017). Left frontal pole theta burst stimulation decreases orbitofrontal and insula activity in cocaine users and alcohol users. Drug and Alcohol Dependence.

This study compared cTBS to frontal pole to a sham stimulation of muscles with electrodes. The study found that participants could not distinguish TMS from sham stimulation. More importantly for our purposes they also found that pain subsided quickly “Subjective reports indicated that the painfulness of the protocol subsided after the first 15-30 s”.

The following other studies uncovered by the same search did not report measures of pain / discomfort.

Costa, A., Oliveri, M., Barban, F., Torriero, S., Salerno, S., Lo Gerfo, E.,.[…] and Carlesimo, G. A. (2011). Keeping memory for intentions: a cTBS investigation of the frontopolar cortex. *Cerebral cortex, 21*(12), 2696-2703.

Costa, A., Oliveri, M., Barban, F., Bonnì, S., Koch, G., Caltagirone, C., and Carlesimo, G. A. (2013). The right frontopolar cortex is involved in visual-spatial prospective memory. PLoS One, 8(2), e56039.

Rahnev, D., Nee, D. E., Riddle, J., Larson, A. S., and D’Esposito, M. (2016). Causal evidence for frontal cortex organization for perceptual decision making. Proceedings of the National Academy of Sciences, 113(21), 6059-6064.

3) The trend of an effect of RFPC stimulation on the information bonus for horizon 1, although smaller than that of horizon 6, seems problematic for an interpretation purely based on directed exploration, since there is no opportunity to exploit the newly acquired information for horizon 1. The authors suggest subjects may become less information-seeking in both conditions (consistent with risk or ambiguity aversion in horizon 1 and reduced directed exploration in horizon 6), but this begs the question of what process or mechanism underlies this decrease in both horizons. Given the broader literature on the role of FPC, one interpretation would be that stimulation has disrupted the FPC's ability to faithfully encode the parameters of a "pending" option that they may choose in the future (e.g. Koechlin and Hyafil, Science, 2007) – in this task this could be seen as the option that has not been selected as frequently or attended to recently during forced choices. However I am sure there are other plausible interpretations. How do the authors interpret this effect across horizons in the unequal condition with respect to the broader literature on FPC?

We have dug into this point more and can now include more detail. What we believe is going on here is a tradeoff between the mean of the prior, *R_0_*, and the information bonus *A*. While this tradeoff does not affect our conclusions that RFPC stimulation selectively affects directed exploration, we believe that the tradeoff does suggest caution when interpreting the horizon 1 result.

In particular, note that in Figure 4, in addition to the information bonus going down in both horizons, the prior mean goes up suggesting a possible tradeoff between the information bonus parameter and the mean of the prior. Such a tradeoff is to be expected in this task because the prior has a larger effect on the more uncertain option – i.e. the option chosen once in the [1 3] condition. This larger effect of the prior means that increasing *R_0_* can have a similar effect to an information bonus in the task by increasing the relative value of the uncertain option (in RL terms, this would be exploration by optimistic initialization).Thus, in the context of this task, the model contains an inherent tradeoff between the information bonus and mean of the prior.

In practice, the tradeoff between *R_0_*and *A* shows up as correlations in the posteriors. This is shown in the updated Figure 5 in the manuscript where we plot samples from the posterior over the change in *R_0_* between stimulation conditions (*R_0_* (vertex) – *R_0_* (RFPC)) against the change in information bonus (*A*(vertex) – *A*(RFPC)). In *both* horizon 1 (panel A) and horizon 6 (panel B) there is a tradeoff between the two parameters. However, while the tradeoff affects the interpretation of the horizon 1 and horizon 6 result *alone*, it does not affect the interpretation of the horizon-based *change* in information bonus (panel C).

In addition to including this new figure, we have addressed this point in the manuscript with the following text:

“In addition to the effect on the information bonus in horizon 6, there is also a hint of an effect on the information bonus in horizon 1 (85% samples less than zero) and on the prior mean R0 (88% samples above zero). […] Taken together, this suggests that we can be fairly confident in our claim that RFPC stimulation has a selective effect on directed exploration.”